# Single cell multiomic analysis reveals diabetes-associated β-cell heterogeneity driven by *HNF1A*

Chen Weng[1,2,8], Anniya Gu[1,3,8], Shanshan Zhang[1,2,8], Leina Lu[1], Luxin Ke ![ORCID][1,2], Peidong Gao[1], Xiaoxiao Liu ![ORCID][1], Yuntong Wang ![ORCID][1], Peinan Hu[1,2], Dylan Plummer[4], Elise MacDonald ![ORCID][1], Saixian Zhang[1], Jiajia Xi[1], Sisi Lai ![ORCID][1,2], Konstantin Leskov ![ORCID][1], Kyle Yuan[1,5], Fulai Jin ![ORCID][1,4,6,7] ✉ & Yan Li ![ORCID][1] ✉

Broad heterogeneity in pancreatic β-cell function and morphology has been widely reported. However, determining which components of this cellular heterogeneity serve a diabetes-relevant function remains challenging. Here, we integrate single-cell transcriptome, single-nuclei chromatin accessibility, and cell-type specific 3D genome profiles from human islets and identify Type II Diabetes (T2D)-associated β-cell heterogeneity at both transcriptomic and epigenomic levels. We develop a computational method to explicitly dissect the intra-donor and inter-donor heterogeneity between single β-cells, which reflect distinct mechanisms of T2D pathogenesis. Integrative transcriptomic and epigenomic analysis identifies *HNF1A* as a principal driver of intra-donor heterogeneity between β-cells from the same donors; *HNF1A* expression is also reduced in β-cells from T2D donors. Interestingly, *HNF1A* activity in single β-cells is significantly associated with lower Na+ currents and we nominate a *HNF1A* target, *FXYD2*, as the primary mitigator. Our study demonstrates the value of investigating disease-associated single-cell heterogeneity and provides new insights into the pathogenesis of T2D.

Loss of β cell mass and impaired β cell function are key mechanisms leading to type I diabetes (T1D) and type II diabetes (T2D)[1,2]. Broad heterogeneity in β cell function and morphology has been reported both within and between islets from the same individual[3–8]. Although unproven, an attractive hypothesis is that β-cell heterogeneity may play a role in diabetes pathogenesis. Presumably, single cell genomic technologies will provide the much-needed tools to investigate β-cell heterogeneity. In the past few years, multiple studies have used single cell RNA-seq (scRNA-seq) and single nuclei ATAC-seq (snATAC-seq) to map cell type-specific transcriptomes and epigenomes in pancreatic islets[9–16]. However, although some scRNA-seq studies have indeed examined β-cell heterogeneity and reported β-cell subpopulations[9,11,13], marker genes defined in these studies do not overlap. More importantly, little is known about the common pathways or factors that govern β-cell heterogeneity, and it remains unclear if β-cell heterogeneity contributes to diabetes.

Several scRNA-seq studies, including our own, mapped T2D signature genes in pancreatic endocrine cell types[9,10,16]. One common

[1]Department of Genetics and Genome Sciences, School of Medicine, Case Western Reserve University, Cleveland, OH 44106, USA. [2]The Biomedical Sciences Training Program (BSTP), School of Medicine, Case Western Reserve University, Cleveland, OH 44106, USA. [3]Medical Scientist Training Program (MSTP), School of Medicine, Case Western Reserve University, Cleveland, OH 44106, USA. [4]Department of Computer and Data Sciences, School of Engineering, Case Western Reserve University, Cleveland, OH 44106, USA. [5]Department of Biochemistry, School of Medicine, Case Western Reserve University, Cleveland, OH 44106, USA. [6]Case Comprehensive Cancer Center, Case Western Reserve University, Cleveland, OH 44106, USA. [7]Department of Population and Quantitative Health Sciences, School of Medicine, Case Western Reserve University, Cleveland, OH 44106, USA. [8]These authors contributed equally: Chen Weng, Anniya Gu, Shanshan Zhang. ✉e-mail: fxj45@case.edu; yxl1379@case.edu

conclusion from these studies is the lack of discrete T2D-specific β-cell subpopulations. Instead, the differences between health and disease are subtle and often masked by individual variation. Due to this reason, conventional statistical methods lack the power to identify disease genes, especially since all past studies only analyzed a handful of T2D donors. Inspired by the successes of single cell trajectory analyses in developmental biology[17,18], we previously developed a regression-based approach (RePACT) to improve the sensitivity to identify disease genes[10]. The key assumption underlying the high sensitivity is that among the β-cells from each donor, there is still heterogeneity relevant to T2D[10]. We demonstrated that even with a small number of donors, RePACT can still identify T2D signature genes in β-cells, many of which have insulin-regulatory functions[10]. However, diabetes-relevant β-cell heterogeneity within the same donor has not been explicitly characterized yet.

Here we aim to directly characterize β-cell heterogeneity with single-cell genomic data and seek evidence for its relevance to diabetes. Because scRNA-seq data is sparse, a major challenge to achieve this goal is to determine whether the observed transcriptome variation represents real cellular heterogeneity or is due to random transcript dropout. To address this problem, we developed a new RePACT-based strategy to connect disease-driven β-cell variation to the single-cell heterogeneity within each individual. We also reason that orthogonal β-cell snATAC-seq data can provide a key validation at the epigenetic level. Furthermore, integration of scRNA-seq, snATAC-seq, Patch-seq, and 3D genome data will allow us to reveal the mechanism and function of β-cell heterogeneity. Our results demonstrate that a better understanding of β-cell heterogeneity may create novel therapeutic opportunities for diabetes.

## Results

### Both scRNA-Seq and snATAC-seq define human islet cell types

We generated 20,437 scRNA-seq (via Drop-Seq[19]) and 37,200 snATAC-seq[20] data using fresh human islets from 7 healthy and 4 T2D donors (Fig. 1a, b, Supplementary Fig. 1, Methods). To ensure robust comparative analysis of endocrine cells between different individuals, we applied a doublets-filtering pipeline for scRNA-seq data based on hormone gene expression and filtered out possible doublet nuclei from the snATAC-seq data using a weighted k-nearest neighbors (KNN)-based approach (Supplementary Fig. 1a, b, Methods). We next used canonical correlation analysis (CCA) co-embedding followed by support vector machine (SVM) classification[21] to simultaneously cluster the clean scRNA-seq cells and snATAC-seq cells into 4 endocrine cell types (α, β, δ, and PP) and 4 non-endocrine cell types (duct, pancreatic stellate cells, acinar, and endothelial cells, Fig. 1c, d) (Methods). Most cells (~80%) are endocrine cell types according to both scRNA-seq and snATAC-seq data (Fig. 1g).

We defined 4353 differentially expressed genes across all cell types which are consistent with our previous reports (Supplementary Fig. 1c, Supplementary Data 1)[10]. After clustering, the aggregated pseudo-bulk ATAC tracks show cell type-specific chromatin accessibility at the expected marker genes (Fig. 1f). We also confirmed that the snATAC-seq data can reveal key lineage-specific transcriptional factors (TFs) activated in different cell populations. For example, NEUROD motif is enriched in all four endocrine populations; PDX motif is enriched in β and δ cells; EHF and RUNX motifs are enriched in duct/acinar cells and PSC cells, respectively[22–26] (Fig. 1c, e). We defined 10,137 endocrine-specific peaks and 21,925 non-endocrine specific ATAC peaks (Fig. 2a); most of these variable peaks are located at intronic and intergenic regions (Fig. 2b), suggesting enhancer functions.

We further clustered the endocrine- and non-endocrine ATAC peaks into 13 clusters based on their cell-type-specificity (Fig. 2c, Supplementary Data 2, Methods). These peaks are predictive on the cell type-specific expression of nearest genes, especially those at the promoter/5'UTR (Supplementary Fig. 2a–c). We observed that many cell type-specific genes have multiple long-distance enhancer peaks

with consistent cell type specificity, such as NEUROD1, SIX3, and IRX1 (Fig. 2d–f, Supplementary Fig. 2d–e). Finally, we scanned the variable peaks (C1 ~ C13) for transcription factor (TF) motif enrichment that are also supported by concordant RNA expression patterns (Fig. 2g, h). As expected, these TFs include many known endocrine TFs, such as MAFA, RFX3, RFX6, NKX6-1, FOXA2, ASCL1, PAX6, etc.

### Multiomic annotation of the specific 3D gene regulome in α- and β-cells

Using a low-input "easy Hi-C" (eHi-C) method[27], we generated Hi-C maps with ~30 K sorted α- and β-cells and performed compartment-, TAD-, and loop-level analyses at 250 kb, 25 kb, and ~5 kb resolution, respectively (Fig. 3a, Supplementary Fig. 3a–b, Methods). At compartment level, there is significant variation between α-cells and β-cells. We identified 310 β-cell-specific and 220 α-cell-specific compartment A regions (euchromatin) (Supplementary Fig. 3c, Methods). Although the cell type-specific A-type compartments are enriched with consistent open chromatin and gene activation (Supplementary Fig. 3d), most of the cell type-specific ATAC-peaks are not present in the compartment switch regions (Supplementary Fig. 3e). At TAD level, the variation between α-cells and β-cells is minimal without significant correlation to cell type-specific transcription or open chromatin (Supplementary Fig. 3a, f–h). We used HiCorr[27] and DeepLoop[28] for loop level Hi-C analyses (Methods). This workflow combines a rigorous Hi-C bias-correction strategy with machine learning-based noise removal, which allows robust identification of chromatin loops from low-depth Hi-C data (Fig. 3a, Supplementary Fig. 3b)[28]. We identified 19,733 β-cell-specific and 17,131 α-cell-specific loop interactions (Fig. 3b, Methods). We found that the cell-type specific loops are significantly enriched with specific ATAC peaks and gene expression (Fig. 3c). Again, although the specific loops are enriched at dynamic compartments, most of the specific loops are present in unchanged compartments (Supplementary Fig. 3i–j).

We next build 3D gene regulation modules to explain cell type-specific gene expression. In both α- and β-cells, >70% cell type-specific genes are marked by specific ATAC peaks at promoters, indicating that transcription initiation is the driving mechanism (Fig. 3d, e). Many specific genes are associated with dynamic long-range epigenetic events and these events appear to be very diverse. We classified the dynamic events into (1) specific loops but enhancers are common (e.g., BAALC and ARNTL2); (2) both loops and enhancers are specific (e.g., OTUD3, LRRTM3, LOXL4, and IRX2) (3) common loops but enhancers are specific (e.g., LUZP6 and SLC14A1). The same genes may associate with multiple types of events but again, only a small number of these loop-level events also overlap compartment changes (e.g., LUZP6 and IRX2) (Fig. 3d–g). Taken together, we conclude that chromatin loops best explain gene transcription and in this study we use loop data to reveal long-range enhancer-promoter gene regulation.

### ScRNA-seq reveals intra-donor T2D-associated β-cell heterogeneity

We previously showed that if we assume the presence of disease-associated β-cell heterogeneity, a trajectory-based algorithm RePACT has much improved sensitivity to identify T2D marker genes from a handful of donors[10] (Fig. 4a). Here we repeated the RePACT analysis with a new cohort of seven healthy and four T2D donors (Fig. 1b). Consistent with our previous results, we observed a continuous transcriptome variation associated with T2D status (Fig. 4b, c). RePACT computed a T2D trajectory and identified 207 up-regulated and 512 downregulated genes from the new cohort; these genes are also consistently changed in our previous Fang et al. cohort (six healthy and three T2D)[10] (Fig. 4d, e, Supplementary Data 3), supporting the robustness of this T2D trajectory.

In the RePACT analysis, cells from the same donor may have different "T2D pseudo indexes" in the T2D trajectory (Fig. 4c), this allows

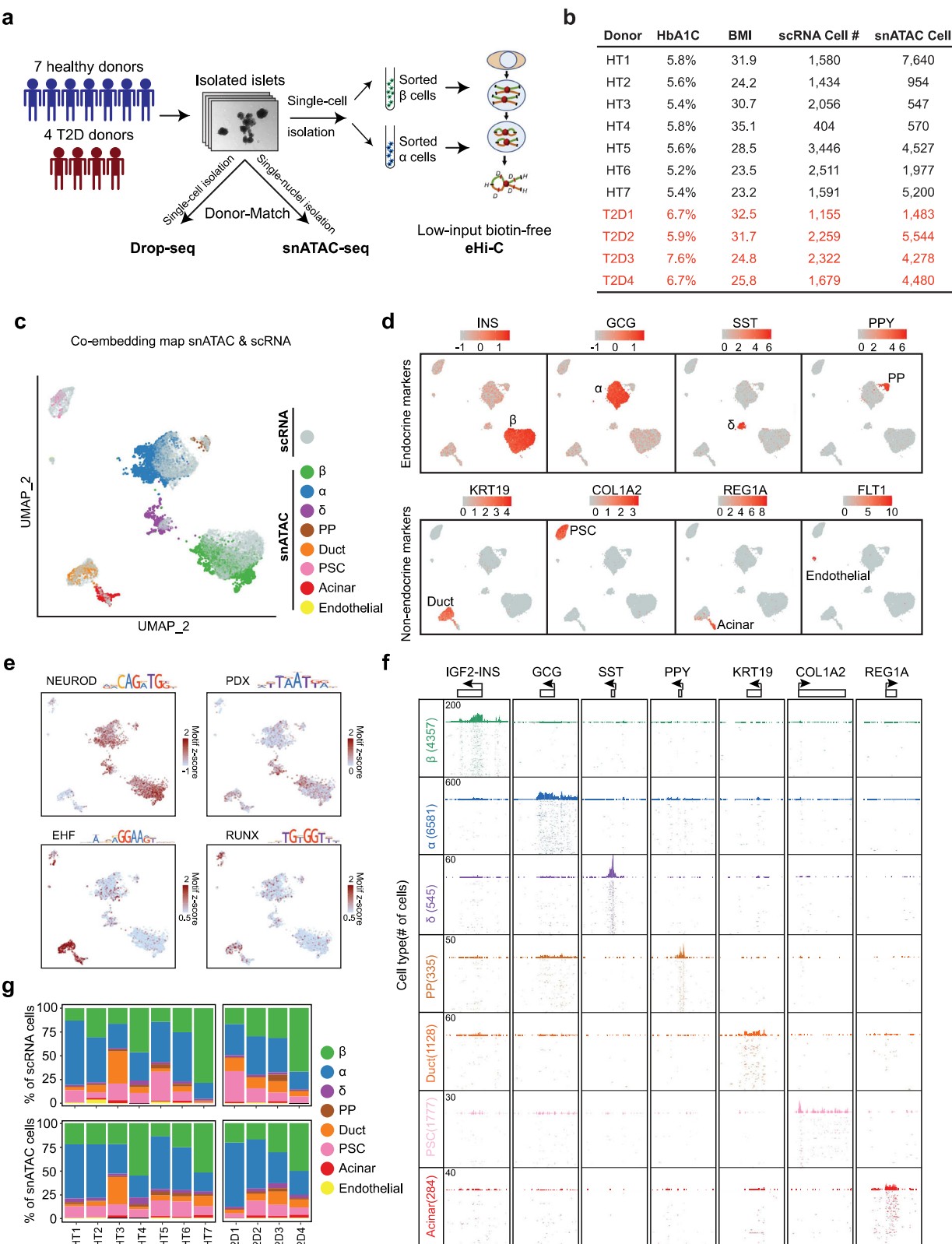

**b**

| Donor | HbA1C | BMI | scRNA Cell # | snATAC Cell # |
|---|---|---|---|---|
| HT1 | 5.8% | 31.9 | 1,580 | 7,640 |
| HT2 | 5.6% | 24.2 | 1,434 | 954 |
| HT3 | 5.4% | 30.7 | 2,056 | 547 |
| HT4 | 5.8% | 35.1 | 404 | 570 |
| HT5 | 5.6% | 28.5 | 3,446 | 4,527 |
| HT6 | 5.2% | 23.5 | 2,511 | 1,977 |
| HT7 | 5.4% | 23.2 | 1,591 | 5,200 |
| T2D1 | 6.7% | 32.5 | 1,155 | 1,483 |
| T2D2 | 5.9% | 31.7 | 2,259 | 5,544 |
| T2D3 | 7.6% | 24.8 | 2,322 | 4,278 |
| T2D4 | 6.7% | 25.8 | 1,679 | 4,480 |

us to directly test if any T2D signature gene is also variably expressed among cells from the same donors, thus revealing the source of T2D-associated β-cell heterogeneity. For example, after grouping β-cells from each donor into quartiles based on their T2D pseudo indexes, downregulated T2D marker genes S*100A10*, *FXYD2*, and *PPP1R1A* and upregulated T2D marker genes *CDKN2A*, *PDE4B*, and *SIX3* all demonstrated a clear trend of heterogeneity within the same donor (Fig. 4f, h),

in both healthy and T2D donors. In contrast, the expression of T2D marker genes *RPL36AL* and *FOS* differ between healthy and T2D donors but show little intra-donor variation along the T2D trajectory (Fig. 4g, i). Based on these observations, we used Fisher's method to systematically classify all T2D marker genes into "intra-donor" and "inter-donor" heterogeneous groups (Fig. 4j, Methods). Interestingly, the "intra-donor" heterogeneous T2D marker genes are

**Fig. 1 | A single-cell multiomic atlas of human islets of Langerhans. a** Schematic of the overall experimental design. Isolated islets from 11 human donors were subjected to donor-matched Drop-Seq and snATAC-seq, as well as Hi-C for sorted β and α cells. **b** Summary of key donor information. **c–e** Unsupervised clustering of both single nuclei from ATAC-seq and single cells from RNA-seq in the same UMAP space for human islets using CCA-based co-embedding method. In (**c**), both RNA and ATAC datapoints are shown, with the highlights of the cell-type identities assigned to each nucleus of ATAC. In (**d**), where only the RNA datapoints are shown,

RNA expression levels of signature genes and the cell-type identifications are visualized on each single cell of RNA. In (**e**), where only the ATAC datapoints are shown, key transcription factor motif frequencies are shown on each nucleus of ATAC. **f** Genome browser snapshots for aggregated pseudo-bulk ATAC tracks of different cell types. 7 signature loci are shown. Each track (row) shows aggregated peaks from single nuclei of each clustered cell type. **g** Summary of cell-type composition in each donor from both the Drop-Seq (scRNA) and the snATAC.

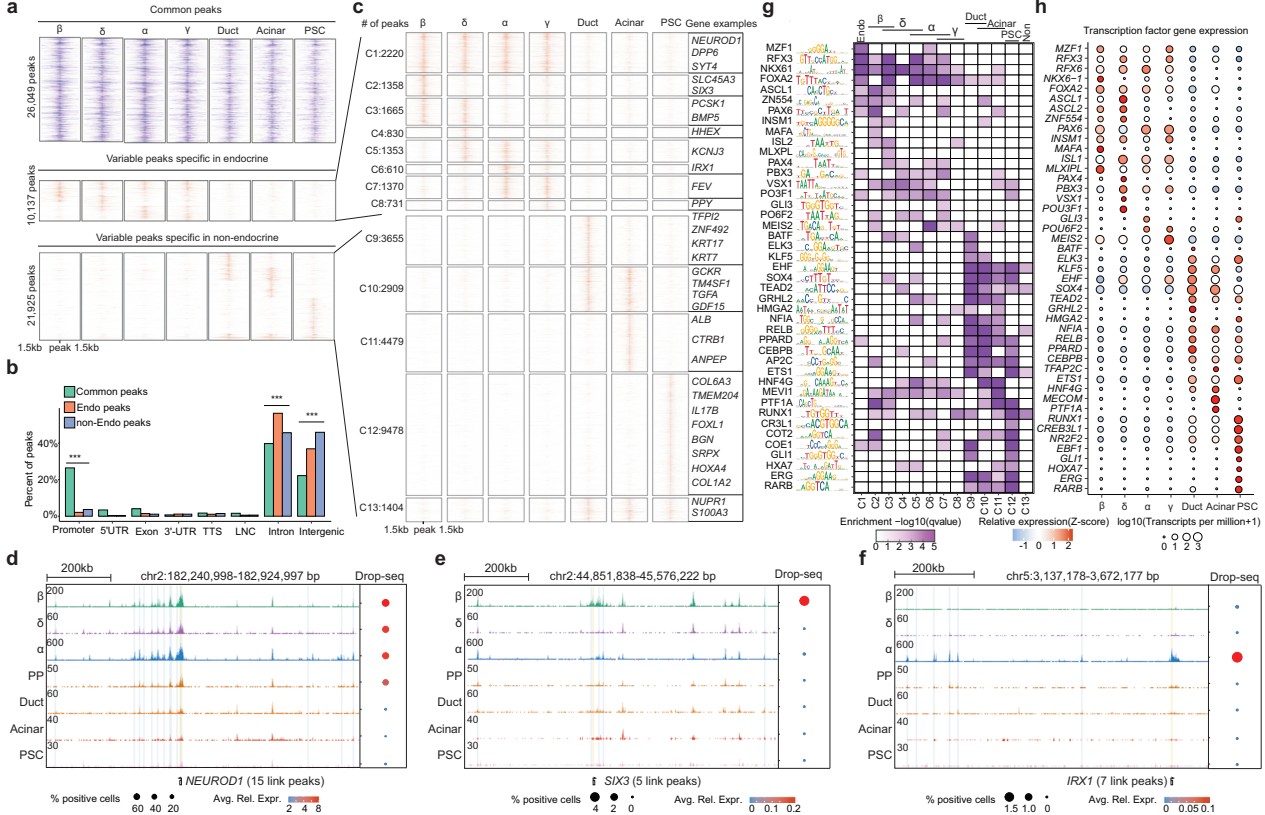

**Fig. 2 | Human islet cell-type-specific chromatin accessibility landscape.**
**a** Heatmap displaying 58,111 open chromatin peaks that are accessible in at least one of the 7 islet cell types. All peaks are classified into three groups: common peaks that are open in all 7 cell types; endocrine-specific peaks, and non-endocrine-specific peaks. 3 kb around the open chromatin peaks are displayed. **b** Percentage of peaks located in each chromatin category: Promoters, 5′- untranslated regions (5′-UTRs), Exons, 3′-untranslated regions (3′-UTRs), Transcription termination sites (TTSs), LincRNAs (LNC), Intron, and intergenic regions. ***p values < 0.001, two-sided Fisher's exact test. **c** k-means clustering for endocrine-specific peaks(C1 - C8) and non-endocrine-specific peaks(C9 - C13). Example genes with promoter peaks

of different clusters are shown. **d–f** Genome browser snapshot of example loci of specific genes. Each track (row) shows aggregated peaks from single nuclei of each clustered cell type. On the right is the bubble plot that shows the aggregated RNA expression level (Drop-Seq Data) for each cell type. Avg.Rel.Expr (color gradient) indicates the average relative expression. % positive cells (bubble size) indicate the percentage of cells that show non-zero expression of the given gene. Three gene loci are shown: *NEUROD1* (**d**), *SIX3* (**e**), *IRX1* (**f**). **g**, **h** Cell-type specific transcription factor activity. The transcription factor motif enrichment in each of the 13 peak clusters in (**g**), and the corresponding transcription factor gene expression level by Drop-Seq in (**h**).

enriched with terms relevant to insulin secretion, stimulus response, and cell proliferation, while the "inter-donor" T2D marker genes are enriched for housekeeping terms such as mitochondria and ribosomal functions (Fig. 4k). These results suggest that intra-donor heterogeneity and inter-donor variation may play different roles in T2D pathogenesis.

**SnATAC-RePACT reveals T2D epigenome heterogeneity in β-cells**

With snATAC-seq data in matched donors, we can investigate if the β-cell heterogeneity is encoded at the epigenetic level. We firstly identify T2D associated dynamic ATAC-peaks by extending the RePACT trajectory analysis to snATAC-seq data with latent semantic indexing (LSI)[29] analysis (Methods). We identified 4623 and 5359 peaks that gain

or lose chromatin accessibility in T2D (Fig. 5a–c, Supplementary Data 3); the dynamics of these peaks can be visualized after grouping all β-cells into bins based on their T2D pseudo index, such as *CDKN2A/B* and *HNF1A* (Fig. 5c, f, g). We used chromVAR[30] to identify TF motifs that are enriched or depleted along the T2D trajectory; these results were then cross-referenced to the scRNA-RePACT results to predict causal TFs (Fig. 5d, e). Notable downregulated TFs include *HNF1A/B* and *RFX6*, which have known diabetes association and/or play important roles in β-cell function[31–35]. Notable up-regulated TFs include *NEUROD1*, nuclear transcription factor (NFYs), and *TP53*, etc. (Fig. 5d, e), which may reflect the β-cell death and dedifferentiation reported in diabetes[36,37].

We next test if the T2D-associated ATAC peaks also have intra-donor heterogeneity. Again, using Fisher's method (Methods), we

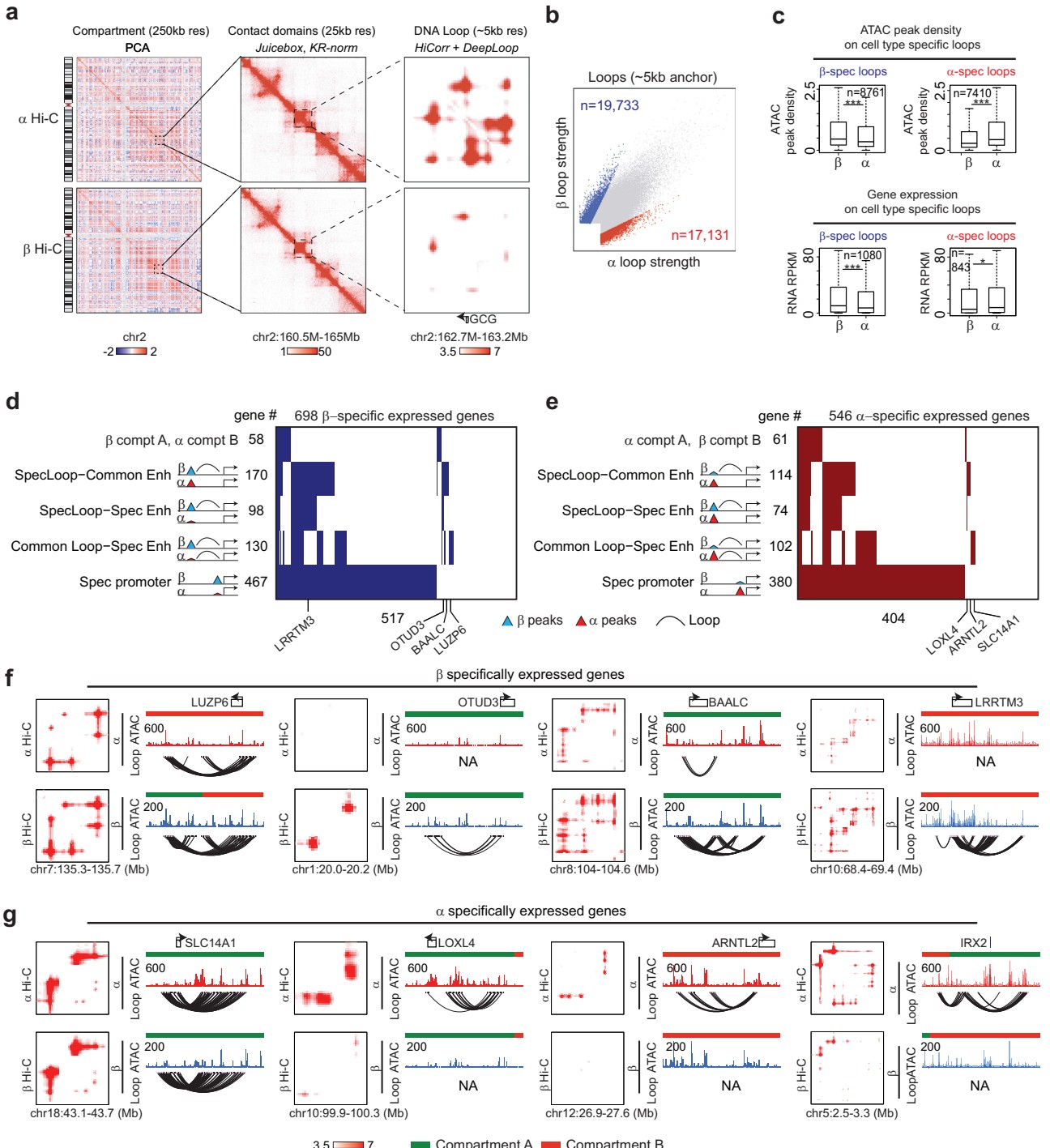

**Fig. 3 | 3D genome organization in islet β and α cell specification. a** Hierarchical genome architecture (compartment, contact domains and DNA Loops) near *GCG* locus in both α- and β-cells. **b** Scatterplot comparing the loop strength in α- and β-cells from DeepLoop analysis. Highlighted are the α- (n = 17,131) or β-cell (n = 19,733) specific loops. **c** The boxplots (median ± upper and lower quartiles) are showing the ATAC (n = 8761) intensity (top) and gene expression (n = 1080) (bottom) for regions with β-cell specific loops (left) and α-cell specific loops (right) (7410 ATAC peaks, and 843 genes). Statistical significance determined via two-sided Wilcox Test. *p-value < 0.05, **p-value < 0.005, ***p-value < 0.0005, upper and lower limits of boxes indicate interquartile ranges, center lines indicate median values, whiskers indicate values with a maximum of 1.5 times the interquartile range and outliers indicate values beyond 1.5 times the interquartile range. **d** The heatmap for 698 β-cell specifically expressed genes. Rows are categories of specificity. **e** Similar to (**d**), the heatmap for 546 α-cell specifically expressed genes. **f** Examples of β-cell-specific genes associated with β-cell-specific loops. Each locus includes contact heatmaps in α-cell and β-cell (left), the tracks for genes, compartments, ATAC and loop curves (right). **g** Similar to (**f**), examples of α-cell-specific genes with α-cell-specific loops.

defined 1962 downregulated and 1602 T2D upregulated ATAC peaks that show "intra-donor" heterogeneity (Fig. 5h). Two examples at *CDKN2A* and *HNF1A* promoters are shown in Fig. 5i, j, in which we show four ATAC-seq tracks for each donor after grouping the β-cells into quartiles based on their T2D pseudo-index. Importantly, many "intra-donor" heterogeneous peaks are proximal to "intra-donor" hetero-geneous genes from the scRNA-RePACT analysis, including down-regulated genes *HNF1A*, *A1CF* and up-regulated genes *CDKN2A*, *SIX3*

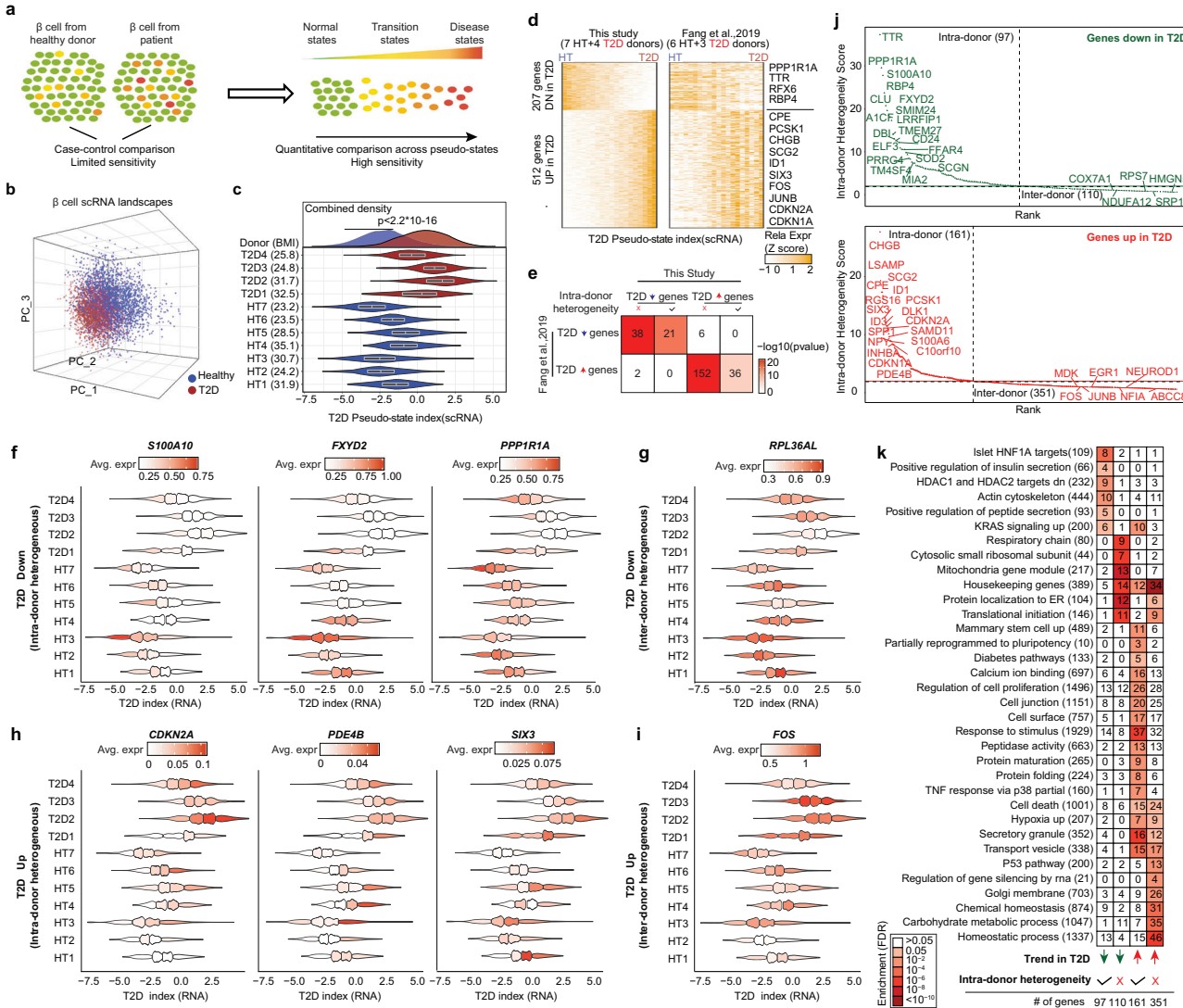

**Fig. 4 | RNA-RePACT dissects the T2D-associated transcriptomic heterogeneity in β-cells. a** Schematics showing the idea of RePACT. A trajectory-based method can improve the sensitivity of disease gene identification if a disease-associated β-cell heterogeneity exists. **b** Plot all β-cells from the scRNA-seq data in the top 3 principal components spaces. Blue or red color indicates whether the cells are from healthy or T2D donors. **c** After building a β-cell T2D trajectory with RePACT using PC1-PC10 (Methods), the violin plot (median ± upper and lower quartiles) compares the distribution of T2D pseudo-index for β-cells from each donor. Density plot (top) compares all β-cells from healthy (n = 7) or T2D donors (n = 4) with p-value (two-sided Kolmogorov-Smirnov test). **d** Left heatmap: the expression changes of the top up/downregulated genes along the T2D trajectory. Right heatmap: the expression changes of the same genes in a previously published cohort of β-cells. **e** reproducibility of intra-donor or inter-donor heterogeneity genes from this study compared to previously published study. Color intensity indicates the odds ratio. *P* values (two-sided Fisher's exact test) and the numbers of genes are shown in the squares. **f–i** Examples of T2D trajectory genes with intra- and inter-donor heterogeneity. Each violin plot still shows the distribution of T2D index of β-cells in all

donors in the same way as (**c**) but with additional information on the expression of denoted gene. In every violin, cells are binned into 4 subpopulation-quartiles according to the T2D index. Normalized gene expression is visualized in each violin quartile. Color scale indicates averaged expression level. Examples of four categories of genes are: **f** T2D-downregulated intra-donor heterogeneous genes; **g** T2D-downregulated inter-donor heterogeneous genes; **h** T2D-up-regulated intra-donor heterogeneous genes; **i** T2D-up-regulated inter-donor heterogeneous genes. **j** Ranking genes based on if they show consistent intra-donor heterogeneity. *Y*-axis indicates the significance from Fisher's method after integrating *p*-values (two-sided *t*-test) from RePACT analyses of every individual donor using T2D trajectory (Methods). T2D-down intra-donor heterogeneous genes (n = 97); T2D-down inter-donor heterogeneous genes (n = 110); T2D-up intra-donor heterogeneous genes (n = 161); T2D-up inter-donor heterogeneous genes (n = 351). Dash lines indicate a cutoff of 2 (integrated *p*-value < 0.01) to classify the genes, examples in (**f–i**). **k** GSEA analysis for the four categories of T2D trajectory genes. Total number of genes of each function term is shown by each row, and the gene number in a tested category is shown in the square.

(Figs. 4j, 5h), therefore providing an orthogonal validation for the transcriptomic intra-donor heterogeneity.

We further use FIMO[38] to scan TF motifs that are enriched in the four categories of T2D trajectory peaks (down- and upregulated peaks, with intra-donor or inter-donor heterogeneity, Fig. 5k, Methods, Supplementary Data 4). Notably, HNF1A motif is enriched within the "intra-donor down" ATAC-peaks (Fig. 5k), agreeing with Fig. 4k that islet *HNF1A* targets are enriched among "intra-donor down" genes from the

scRNA-RePACT analysis. Similarly, p53 motifs are enriched in the "inter-donor up" ATAC-peaks (Fig. 5k), and p53 pathway genes are enriched among "inter-donor up" genes (Fig. 4k).

We finally integrated the scRNA-seq, snATAC-seq, and β-cell Hi-C data to reconstruct regulatory circuits between T2D marker genes and ATAC-peaks. Among all the T2D trajectory genes, we identified 68 T2D-down and 154 T2D-up genes that can be explained by at least one T2D trajectory peak at the promoter or distal regions connected by Hi-C

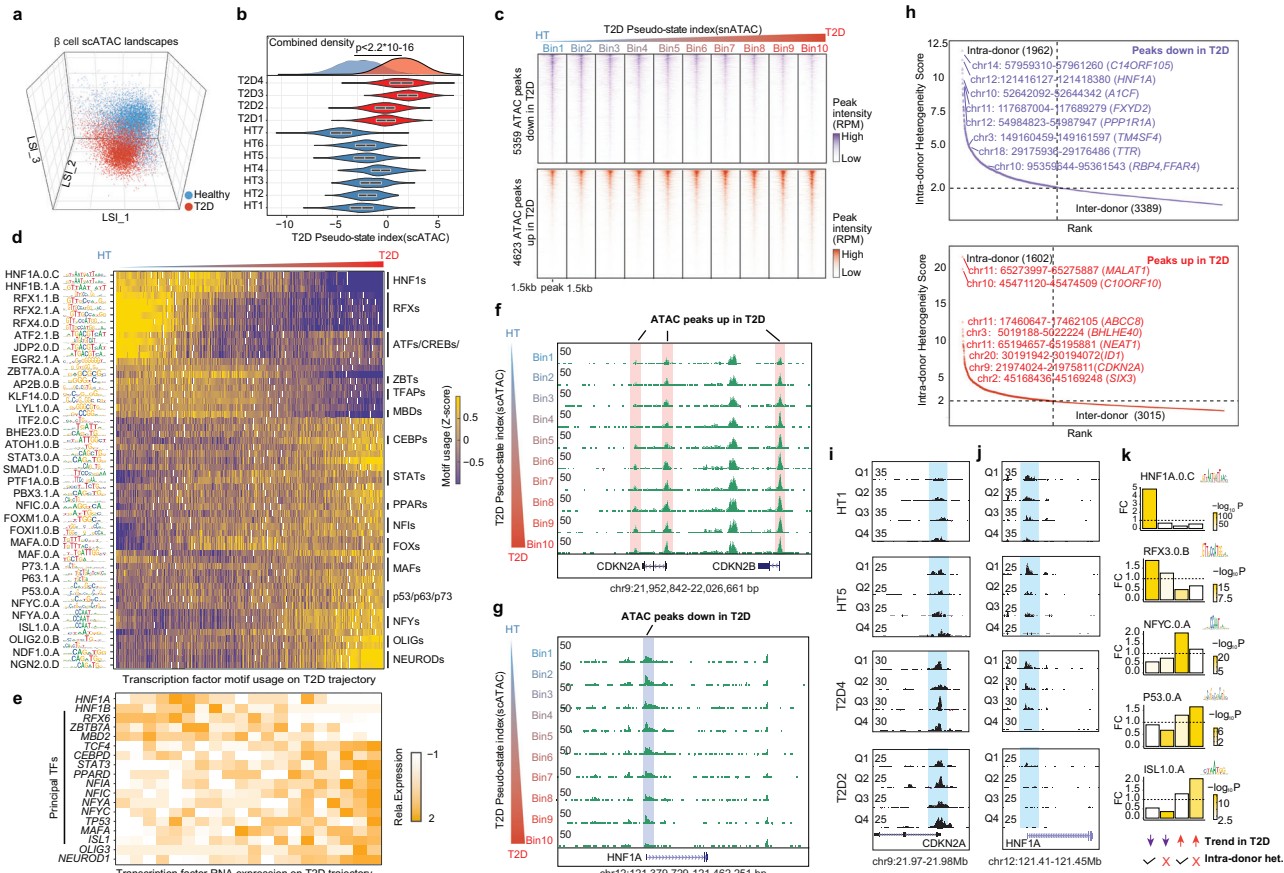

**Fig. 5 | ATAC-RePACT dissects the T2D-associated epigenomic heterogeneity in β-cells. a** Plot all β-cells from snATAC-seq data in the top 3 latent semantic indexing (LSI) spaces. Cells are colored based on whether the donor is healthy or T2D. **b** Like Fig. 4c, compare the distribution of the T2D pseudo-state index derived from snATAC (T2D trajectory-snATAC) of each donor (7 healthy donors, 4 T2D donors). *P* value is from two-sided Kolmogorov-Smirnov test. The T2D trajectory (snATAC) is computed from the regression analysis using LSI1-LSI10 (Methods). Violin plots show median ± upper and lower quartiles. (Two sided Kolmogorov-Smirnov test was performed to calculate *p*-value) **c** Heatmap displaying the ATAC peaks that are variable along the T2D trajectory. All β cells are binned into 10 bins along the T2D trajectory for visualization. 3 kb around the open chromatin peaks are displayed. (**d**) Transcription factor motif usage (Z-score) along the T2D trajectory in β-cells. The motif usage is computed at single-cell level with chromVAR (Methods). Selected motifs are shown. **e** Transcription factor gene expression change on T2D trajectory from RNA-RePACT analysis. **f**, **g** Genome browser snapshots of peaks that

are losing or gaining accessibility along the T2D trajectory. Single cells from snATAC-seq data are grouped into 10 bins along the T2D trajectory. Each track (row) is a pseudo-bulk ATAC-seq track of one bin of cells. *CDKN2A* gains ATAC peak (**f**) and *HNF1A* loses ATAC peak (**g**). **h** Like in Fig. 4j, we also rank T2D-regulated ATAC peaks based on their scores of intra-donor heterogeneity. Dashed lines indicate the cutoff to define four categories of peaks: T2D-down intra-donor heterogeneous peaks (n = 1962); T2D-down inter-donor heterogeneous peaks (n = 3389); T2D-up intra-donor heterogeneous peaks (n = 1602); T2D-up inter-donor heterogeneous peaks (n = 3015). **i**, **j** Example of the intra-donor heterogeneous peak at *CDKN2A* and *HNF1A* promoters. For each donor, the β cells from snATAC-seq data are binned into four quartile subpopulations according to the ATAC T2D-pseudo-index; each panel shows the pseudo bulk ATAC-seq tracks of the four quartiles. **k** Transcription factor motifs enriched in the four categories of ATAC peaks, color of the boxes indicates the significance of enrichment from two-sided Fisher's exact test.

---

loops (size range 41 kb to 1.3 Mb; median size 221 kb) (Fig. 6a, Supplementary Fig. 4). As expected, T2D-down genes are more likely to connect with T2D-down peaks, and T2D-up genes are more likely to connect with T2D-up peaks; the enrichment is more significant between "intra-donor" heterogeneous genes and "intra-donor" heterogeneous ATAC-peaks (Fig. 6b). These results strongly argue that the observed T2D-associated β-cell heterogeneity is encoded in the epigenome and driven by variable transcription programs in individual cells.

### *HNF1A* drives both intra-donor and inter-donor β-cell heterogeneity

Our analyses repeatedly highlight the loss of *HNF1A* activity in the T2D trajectories at both inter-donor (Fig. 5d, e, g) and intra-donor levels (Figs. 4k, 5h, j, k). We therefore looked for T2D trajectory peaks containing HNF1A motifs and used the peak-gene connections from Fig. 6a to predict *HNF1A* downstream genes (Fig. 6c, Methods). For example,

promoter or enhancer peaks at *TTR*, *SMIM6*, and *SPRY1* loci all contain HNF1A motifs and lose accessibility along the T2D trajectory (Fig. 6d, Supplementary Fig. 5a–b); all three are downregulated in T2D and show intra-donor heterogeneity. We also picked predicted *HNF1A* target genes for RT-qPCR validation with *HNF1A* siRNAs in a human β cell line (EndoC-βH3). Two distinct *HNF1A* siRNAs both achieve knockdown efficacy at 30 ~ 40%. Out of the twelve genes tested, eight genes are significantly downregulated by at least one siRNA (Fig. 6e). Finally, we verified with *Cut&Run* in EndoC-βH3 cells that most of the predicted target genes have HNF1A binding sites, and HNF1A colocalizes with the T2D-downregulated ATAC-seq peaks (Supplementary Fig. 6).

We further compared HNF1A levels in β cells from eight cadaveric human islets (four healthy and four diabetic) using two-way flow cytometry analyses after immunostaining with antibodies against both C-peptide and HNF1A. Consistent with genomic results, the HNF1A signal in β-cells (C-pep+) from diabetic donors is significantly lower than those from healthy controls, again suggesting a connection

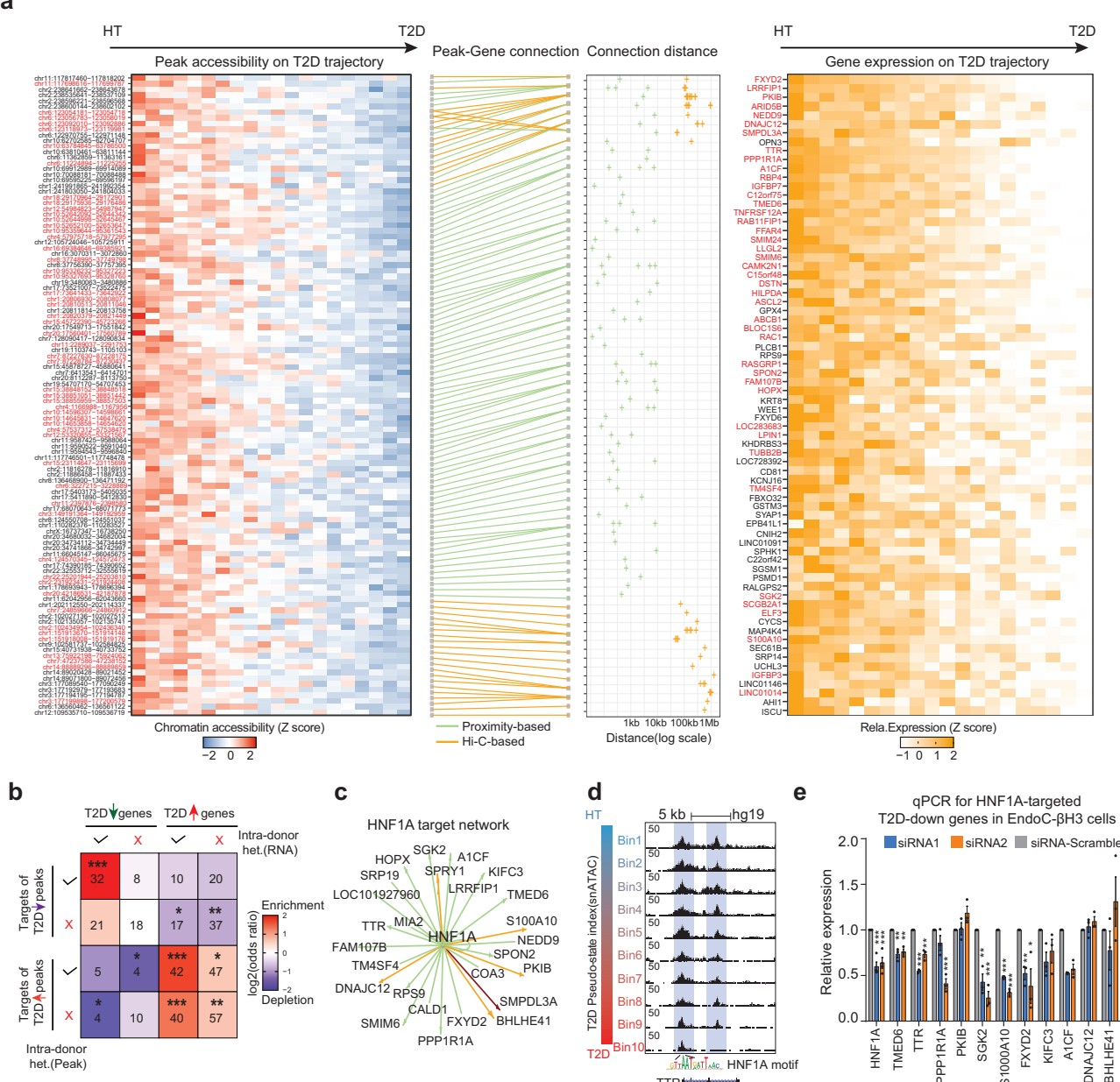

**Fig. 6 | Multiomic reconstruction of the T2D gene regulatory circuits in β-cells.** **a** Connect T2D downregulated ATAC peaks to downregulated genes from RePACT analyses through multiomic integration (T2D upregulated peak-gene pair in Supplementary Fig. 4). Leftmost panel: heatmap of peaks losing accessibility along T2D trajectory; peak locations (hg19) are on the left of each row. Peaks highlighted in red show intra-donor heterogeneity. Second panel shows the peak-gene connections. Green lines: peaks within 10 kb of the TSS; orange lines: distal peak-gene pairs supported by Hi-C loops. Third panel: the distance of each peak-TSS connection. Rightmost panel: heatmap of T2D downregulated genes. Red-highlighted gene names indicate intra-donor heterogeneous genes. **b** Enrichment analysis between T2D trajectory genes and T2D-trajectory peaks. Color intensity indicates the odds ratio. P values (two-sided Fisher's exact test) and the numbers of genes are shown in the squares. **c** A subnetwork of predicted T2D downregulated genes controlled by *HNF1A* (Methods). Green lines: peaks within 10 kb of the TSS; orange lines: distal peak-gene pairs supported by Hi-C loops. Maroon: both proximal and distal peaks. **d** Genome browser snapshot of *TTR* locus (chr18:29,169,915-29,180,800) which is a putative *HNF1A* target. **e** qPCR validation of selected *HNF1A* targets following *HNF1A* knock-down. P-values are from two-sided paired *t*-test. ∗∗∗*p* < 0.0005, ∗∗*p* < 0.005, ∗*p* < 0.05 (three biological replicates each, *HNF1A* has four biological replicates. Each biological replicate is the average of three technical replicates). Data are presented as mean values +/− SEM.

between *HNF1A* activity and disease state in β-cells (Fig. 7a, Supplementary Fig. 5c). Consistent with these results, after siRNA knockdown of *HNF1A* in EndoC-βH3 cells, we observed lower overall levels of secreted insulin under both low- and high-glucose conditions (Fig. 7b).

To validate the "intra-donor" heterogeneity of *HNF1A*, we further performed three-way flow cytometry analyses on human islets from healthy donors: each experiment includes antibodies against C-peptide, HNF1A, and a third protein of interest. Despite a general positive

correlation between C-peptide and HNF1A in β-cells, we reproducibly observed β-cells with high HNF1A levels relative to C-peptide levels (Fig. 7c, d, Supplementary Fig. 5d, e, middle panels). Similar β-cells are also observable when we plot HNF1A target genes (TTR, A1CF, PKIB, and TMED6) against C-peptide (Fig. 7c, Supplementary Fig. 5d, right panels). In contrast, none of the control genes (NEUROD1, NKX2-2, and SIX3) show noticeable β-cell deviance in the same analysis (Fig. 7d, Supplementary Fig. 5e, right panels). Importantly, the HNF1A-high cells

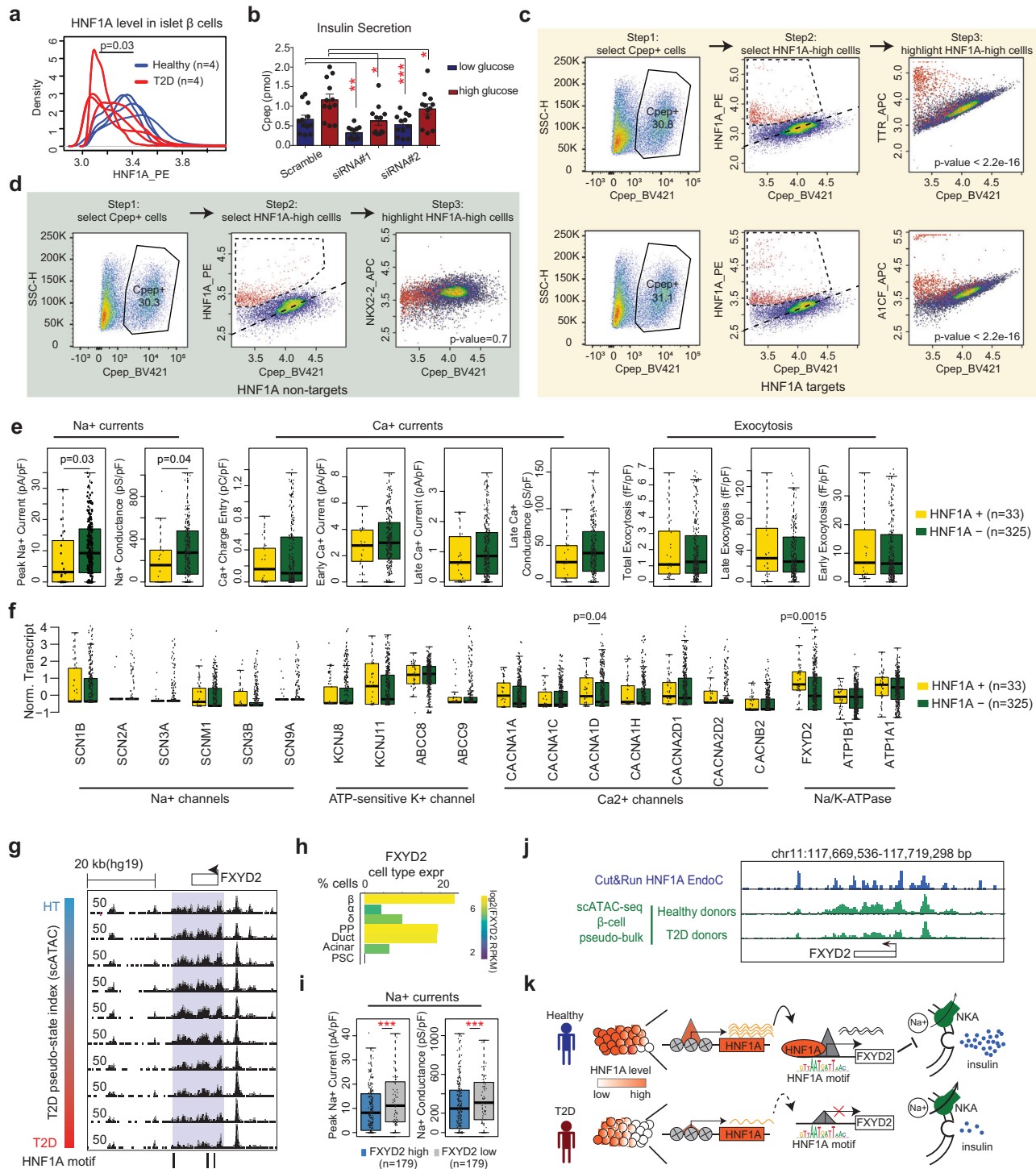

are significantly overrepresented amongst the high TTR, A1CF, PKIB, and TMED6 cells (highlighted cells Fig. 7c and Supplementary Fig. 5d). Taken together, the flow cytometry data are consistent with the genomic results and support β-cell heterogeneity driven by *HNF1A* activity.

### *HNF1A* is linked to reduced sodium influx in β-cell heterogeneity

To further explore the physiological function of *HNF1A* marked β-cell heterogeneity, we analyzed independent human islet Patch-seq data which combines scRNA-seq with electrophysiological measurements of exocytosis and channel activity[39]. Due to the issue of high dropout rate, *HNF1A* transcripts are detectable in only 33 out of 358 β-cells in

this dataset. Nevertheless, the 33 *HNF1A* + β-cells show significantly lower peak Na$^+$ influx, while other electrophysiological properties, including Ca2+ currents and exocytosis activity, are not significantly different (Fig. 7e).

Since the statistical power of Patch-seq analyses are affected by the high dropout rate of *HNF1A* gene transcripts, we reasoned that *HNF1A* target genes may serve as better markers for β-cell heterogeneity. We firstly reanalyzed the transcriptome component of the Patch-seq data and confirmed that 8 (out of 18) putative *HNF1A* target genes express significantly higher in the 33 *HNF1A*+ cells (Supplementary Fig. 7a). Strikingly, nearly all (14 out of 18) the *HNF1A* target genes, but not the control genes, are associated with decreased Na+

**Fig. 7 | Intra-donor heterogeneity of *HNF1A* and its target genes. a** The distribution of HNF1A level from flow cytometry analyses in β cells from four healthy islets (blue) and four T2D islets (red). Statistical significance determined via two-sided Wilcox test. **b** Glucose stimulated insulin secretion in *HNF1A* siRNA knock-down EndoC-βH3 cells or scrambled siRNA control cells. C-peptide levels in low glucose (2.8 mM) and high glucose (20 mM) conditions. Statistical significance was determined using paired two-sided *t*-test from *n* = 12 wells collected from four separate experiments by comparing corresponding glucose conditions to each other. *\*p*-value < 0.05, \*\**p*-value < 0.005, \*\*\**p*-value < 0.0005. Data are presented as mean values +/− SEM. **c, d** Three-way flow cytometry analysis of β-cells. Left panel: we first select the β-cells (C-pep+). Middle panel: HNF1A-high β-cells (highlighted in red) can be determined from the HNF1A vs. C-pep scatter plots. Right panel: for the β-cell population, we plot the gene of interest against C-pep in a scatter plot, HNF1A-high cells are highlighted in red. Examples of *HNF1A* target genes TTR and A1CF are shown in (**c**); Example of *HNF1A* non-target gene NKX2-2 is shown in (**d**). In the right panels, one sided *t.test* is used to measure the difference of target gene expression between HNF1A high cells and the rest of the cells. **e** Compare the electrophysiological measurements of exocytosis and channel activity in *HNF1A* + β cells (*n* = 33) and *HNF1A*- β cells (*n* = 325) from Patch-seq data, *p*-values from two-sided Wilcox test. Boxplots show median ± upper and lower quartiles. **f** Compare channel gene expression in *HNF1A*+ and *HNF1A*- β cells in Patch-seq data (Methods). *p*-value from two-sided Wilcox test. Boxplots show median ± upper and lower quartiles. **g** Browser snapshot of *FXYD2* locus (chr11:117,660,352-117,716,145) with HNF1A motifs indicated. Same as Fig. 5f–g, each track (row) is a pseudo-bulk ATAC track of one bin of cells (out of 10) on the T2D trajectory. **h** *FXYD2* expression in pancreatic islet cell types from the scRNA-seq data. **i** Na+ current measurements in *FXYD2* high (*n* = 179) and *FXYD2* low cells (*n* = 179) from Patch-seq data. \*\*\**p*-value < =0.0005, two-sided Wilcox test. Boxplots show median ± upper and lower quartiles. **j** Genome browser snapshot of *FXYD2* locus with HNF1A Cut&Run (blue) and pseudo-bulk ATAC track in β cells from T2D and healthy donors (green). **k** A schematic to show the role of *HNF1A* in β-cell heterogeneity and T2D. *HNF1A* expression has cell-to-cell variation in β-cells and is lower in T2D. *HNF1A* regulates *FXYD2*, an inhibitor of the Na+/K+-ATPase (NKA). *FXYD2* decreases NKA's affinity for Na+ ions and promotes depolarization and insulin secretion.

---

influx (Supplementary Fig. 7b). Therefore, the Patch-seq data independently validate the putative *HNF1A* target genes and strongly support a connection between *HNF1A* activity and reduced Na+ influx in β-cells.

In patch-clamp experiments, the variable ion current is most frequently attributed to changes in ion channels. To explore why *HNF1A* + β-cells have lower Na+ influx, we examined the Patch-seq data of all detectable ion channel genes. As expected, the fluctuations of several key ion channel genes are associated with variable ion currents. For example, high levels of Na+ channel (*SCN3A*) and ATP-sensitive K+ channels (*KCNJ8* and *ABCC9*) are all associated with high Na+ influx; Ca2+ channel *CACNA1A* is positively correlated with Ca2+ current but not Na+ current (Supplementary Fig. 7c). However, none of the ion channel genes show variable expression between *HNF1A*+ and *HNF1A*-cells except *CACNA1D*, which shows higher expression in *HNF1A*+ cells (Fig. 7f). However, *CACNA1D* heterogeneity is not associated with Na+ activity (Supplementary Fig. 7c) and therefore cannot explain *HNF1A*'s association to the Na+ phenotype.

Interestingly, out of all predicted *HNF1A* target genes, *FXYD2* is the only one with known function to regulate ion transport across the cell membrane (Fig. 6a, c, e). *FXYD2* is highly expressed in β cells (Fig. 7h) and like *HNF1A*, *FXYD2* is downregulated in the T2D trajectory at both transcriptome and epigenome levels with intra-donor heterogeneity (Figs. 4f, 6a, 7g). HNF1A binding at the *FXYD2* locus, corresponds to chromatin accessibility peaks downregulated in T2D (Fig. 7j). *FXYD2* is an inhibitory subunit of the Na+/K+-ATPase[40,41]. In cells, the Na+/K+-ATPase plays a critical role in maintaining ion gradients and membrane resting potential. Therefore, *FXYD2* may lower ion gradients, facilitate membrane depolarization, and presumably reduce the Na+ influx necessary for an action potential. Consistent with this model, Patch-seq data show a significant negative association between *FXYD2* and Na+ current (Fig. 7i, Supplementary Fig. 7c). Conversely, *ATP1A1* (the catalytic subunit of Na+/K+-ATPase) is positively associated with Na+ current (Supplementary Fig. 7c). Taken together, these results support a model that *HNF1A* reduces Na+ current in β-cells by upregulating *FXYD2* (Fig. 7k, see discussion).

## Discussion

The heterogeneity of β-cells has been described with many molecular and behavioral measurements[3–8], but few studies have managed to link β-cell heterogeneity to diabetes pathogenesis. In this study, our trajectory-based scRNA-seq analysis pinpointed a subset of "intra-donor" T2D signature genes that are also variable among cells from the same donors, enabling the genomic characterization of disease-relevant β-cell heterogeneity. We found that in contrast to the "inter-donor" T2D signature genes, "intra-donor" T2D signature genes are enriched with genes relevant to cell cycle regulation and β-cell

functionality. Importantly, because the "intra-donor" gene signature shows variation in cells from both healthy and diabetic donors, a very attractive hypothesis is that these genes mark the transition toward a disease state.

It is interesting that inter-donor heterogeneous T2D genes are enriched with house-keeping functions including mitochondria genes. Mitochondria play a central role in coupling glucose metabolism to insulin release, and mitochondrial metabolism is substantially inhibited in diabetic β-cells[5,42–46]. For example, the respiratory chain genes NADH dehydrogenase (*NDUFA1/A12/B1/B2/B4/B9*), Cytochrome C Oxidase (*COX7A1*), and Ubiquinol-Cytochrome C Reductase (*UQCRQ*) are all downregulated in T2D patients, but largely homogeneous within the cells from individual donors. There are two possible mechanisms: (i) inter-donor heterogeneous genes may represent a pre-existing uniform expression pattern that varies by individual, which might confer different susceptibility to T2D; (ii) alternatively, inter-donor heterogeneous genes may obtain this variation after the onset of disease, and all cells respond to the disease state rather uniformly. Supporting the second possibility, immediate early genes (IEGs) *FOS* and *EGR1* both respond to metabolic stimuli including glucose and cAMP[3,47,48]; they are also both upregulated in T2D without intra-donor heterogeneity.

In single cell analysis, subpopulations can be most reliably identified as discrete cell clusters; we did not observe discrete β-cell clusters from our data. However, β-cell heterogeneity can also be continuous, which may still cause molecular consequences with disease relevance. Conversely, discrete subpopulations are not necessarily disease-relevant even if they exist. With RePACT, we explicitly investigated continuous β-cell heterogeneity within the same donor and identified *HNF1A* as a disease-relevant heterogeneity driver. It remains to be determined whether the continuous heterogeneity is a stable condition or represents dynamic cells transitioning between *HNF1A*-low or *HNF1A*-high states. Regardless, by integrating scRNA-seq, snATAC-seq, and Hi-C data, we identified many *HNF1A* target genes with consistent intra-donor heterogeneity at both transcriptional and epigenetic levels, thus providing important evidence that *HNF1A*-driven intra-donor heterogeneity is indeed functional.

In humans, heterozygous mutations in *HNF1A* are sufficient to cause the most frequent form of maturity onset diabetes of the young (MODY3)[31]. Since most of the *HNF1A* mutations in MODY3 are simple loss-of-function mutations, it has been proposed that the β-cell is particularly vulnerable to decreased *HNF1A* gene dosage[49,50]. Furthermore, both rare and common variants at the *HNF1A* locus have been associated with T2D[51,52]. We also observed compromised insulin secretion in EndoC-βH3 cells after knocking down *HNF1A*, consistent with several reports in mice and stem cell models[53–56]. Taken together, the link to *HNF1A* suggests a causal role of β-cell heterogeneity in T2D pathogenesis.

By leveraging Patch-Seq data, we found an unexpected association of *HNF1A* activity to decreased Na+ influx in β-cells, most likely through upregulating *FXYD2*, a negative modulator of the Na+/K+-ATPase. This model is intriguing because *FXYD2* is the only β-cell differentially expressed gene in T2D that has been convincingly replicated across multiple scRNA-seq studies, regardless of the choice of platform and analytic tools[57]. However, there is a suspicion that *FXYD2* may not be important for β-cell dysfunction in diabetes since it is not genetically associated to T2D[57]. Our discovery of *FXYD2* being a primary target of *HNF1A*-driven β-cell heterogeneity explained its robust detection as a T2D signature gene and supports its disease relevance.

*FXYD2* loss has been shown to hyperpolarize membrane potential in neurons and decrease action potential firing[41]. We propose that a similar mechanism may exist in *HNF1A*-driven β-cell heterogeneity, where partial or progressive loss of *HNF1A* and *FXYD2* causes membrane hyperpolarization and impaired insulin secretion in MODY3 or T2D (Fig. 7k). Interestingly, this model can also explain why *HNF1A*-MODY patients are particularly sensitive to sulfonylureas[58], which block ATP-sensitive K+ channels and promote membrane depolarization. Taken together, our single cell integrated genomic analysis demonstrates the existence of T2D-associated β-cell heterogeneity with a specific electrophysiological trait driven by a *HNF1A/FXYD2* module. Future studies are needed to validate the contribution of the *HNF1A/FXYD2* pathway to diabetes and to explore options to manipulate β-cell heterogeneity for clinical uses.

## Methods

### Ethics declaration
All research was carried out in compliance with the relevant ethical regulations. The research in this study is not considered human subjects research and was deemed exempt from IRB approval by the CWRU IRB.

### Statistics and reproducibility
No statistical method was used to predetermine sample size. No data were excluded from the analyses. The experiments were not randomized. The Investigators were not blinded to allocation during experiments and outcome assessment.

### Experiment
**Tissue culture and experimental validation**
**Human islet culture.** Human islets were purchased from Prodo Laboratories Inc. Upon receiving, the islets were washed twice with complete PIM(S) Prodo Islet Media (Prodo Laboratories Inc, PIM-S001GMP). Islets were then collected and gently resuspended in complete PIM(S) media and cultured in a six-well non-tissue culture-treated plate overnight. To dissociate islets into single cells, cells were washed once in HBSS (Sigma-aldrich, #6648) and incubated in Accutase (Innovative Cell Technologies, #AT104) at 37 °C for 20–25 min. The islets were broken up gently with a 1 ml pipette every 5 min. When >95% of the islets were digested into single cells, PIM(S) medium was added to neutralize the Accutase and the suspension was passed through a 40 µm cell strainer. Single cells were washed again with HBSS and resuspended in HBSS at the concentration of $2 \times 10^5$/mL for Drop-Seq analysis. To collect sorted human β and α cells, the obtained single cells were first fixed and permeabilized with BD cytofix/ cytoperm buffer (BD, Cat# 51-2090KZ) for 20 min. Then, cells were stained with anti-glucagon antibody (BD Biosciences, Cat# 565891) for α cells and anti-insulin antibody (BD Biosciences, Cat# 565689) for β cells before they were sorted on a BD FACS ARIA II cell sorter.

**Cell culture.** EndoC-βH3 cells were acquired from Univercell-Biosolutions. EndoC-βH3 cells were cultured on 10 cm TPP plates (Sigma #Z707686) as described[59]. Plates were first coated with ECM solution (10 ml of DMEM (Gibco #11960044), 100 µl ECM gel (Sigma

#E1270), 20 µl fibronectin (Sigma #F1141), 100 µl Streptomycin-Penicillin (Gibco #15140122), 5 ml per 10 cm dish) for 1 h at 37 °C before use. Cells were cultured in DMEM (Sigma #D6046) with 2% BSA fraction V (Sigma #10775835001), 50 µM 2-mercaptoethanol (Gibco #31350010), 10 mM Nicotinamide (Sigma #481907), 5.5 µg/ml Transferrin (Sigma #T8158), 6.7 ng/ml sodium selenite (Sigma #214485), and Penicillin/Streptomycin (Gibco #15140122). Before use, cells were treated with 1uM tamoxifen (Selleck #S7827) 2x/week for 21 days. Successful excision of the immortalizing transgene was confirmed by observing the cell morphology.

**siRNA knockdown and qPCR.** EndoC-βH3 cells were plated in 12-well plates at ~0.25 M cells/well one day before transfection. siRNAs were purchased from Dharmacon. Nontargeting control (#D-001810-01-20), *HNF1A* siRNA #1 (#D-008215-01-0010), and *HNF1A* siRNA #2 (#D-008215-02-0010) were added at a final concentration of 25 nM with 1ul lipofectamine RNAi Max (Invitrogen #13778030) diluted in opti-MEM (Gibco #31985070). Media was changed 24 h after transfection. At 72 h post-transfection, RNA was harvested using Zymo's Microprep kit (#R1050). cDNA was generated using Applied Biosystems' high-capacity kit (#4368814). Each qPCR reaction contained 20 µl of final volume (1ul input cDNA at ~10 ng/ul, 10 µl Radiant SYBR Green Lo-ROX 2x qPCR Master Mix from Alkali Sci (#QS1050), and 1ul forward and reverse primers from 10 µM stock solutions.) qPCR primer sequences are listed in Supplementary Table 4. All gene values were normalized to *GAPDH* or *actin* and both siRNAs were then normalized to the scrambled control using the ΔΔCt method. Statistics were calculated using a paired *t*-test from three biological replicates with each biological replicate being the average of three technical replicate wells except *HNF1A* which was calculated from four biological replicates.

**Flow cytometry.** Human islets from Prodo Laboratories were cultured and digested into singlets as previously described. After counting, cells were washed with 1× dPBS, fixed using eBioscience's Transcription Factor Staining Set (#00-5523-00) and stored at −80 °C until ready for use. Samples were thawed in blocking solution (5% BSA-PBS + 0.1% Triton X-100) for 1 h on ice. After washing with 1x Perm/Wash buffer, samples were aliquoted into FACS tubes and primary antibody was added at ~1:50 dilution and stained overnight at 4 °C. Unless otherwise noted, each tube was stained for HNF1A, Cpep, and select target gene. All antibodies are listed in Supplementary Table 5. Minus one controls were used to gate for negative populations. Before adding the secondary antibody, samples were washed once with 1× Perm/Wash buffer. Secondary antibody was added at ~1:125 dilution and stained in the dark at room temperature for 20 min. Samples were washed once more with 1× Perm/Wash buffer, resuspended in 1× dPBS, and run on a BD LSR II, FACS ARIA, or FACS ARIA-SORP. Gating of Cpep+ populations was done using Flowjo v10. The scale values of the Cpep+ populations were then exported, and further analysis was performed using R.

**Glucose-stimulated insulin secretion.** 48 h after siRNA transfection, cells were starved for 24 h in 2.8 mM EndoC-βH3 media (basal media Thermofisher #11966025 with supplemented glucose from Gibco #A2494001, and supplements as described above). 72 h post-transfection, cells were then starved for 1 h in 2.8 mM KRB buffer (115 mM NaCl, 5 mM KCl, 1 mM CaCl$_2$, 1 mM MgCl$_2$, 0.2% BSA, 10 mM HEPES, 24 mM NaHCO$_3$, 2.8 mM glucose). After 1 h, cells were washed with 2.8 mM KRB buffer and then cultured in either low glucose (2.8 mM) or high glucose (20 mM) KRB buffer for 1 h. Supernatant was collected and spun down at 800 × *g* for 5 min to pellet cell debris. C-peptide levels were quantified using Mercodia's Ultrasensitive C-pep Eliza Kit (#10-1141-01). Further analysis was generated in Prism v6.

## Library preparation

**Drop-Seq.** We have established an in-house Drop-seq protocol based on previous publications[10,19]. Briefly, three pump-controlled syringes with cell suspension (200,000 cells/mL), barcoded beads in lysis buffer (200,000 beads/mL), and droplet generation oil were connected to a microfluidic device under microscope supervision to generate droplets. Next, the droplets were broken with Perfluoro-1-octanol followed by vigorous shaking and beads were recovered and used to perform 1st strand cDNA synthesis. The resulting full-length cDNA were fragmented and prepared for sequencing. Libraries were sequenced on HiSeq X sequencers with 150 paired end sequencing.

**Transposome generation.** The Tn5 oligo sequences are listed in Supplementary Table 2. All oligos were diluted into 200uM using dilution buffer (10 mM Tris-pH7.5, 50 mM NaCl). 200uM barcode oligos were mixed (1:1) with 200uM eluted pMENT oligos incubated at 95 °C for 3 min on a thermal cycler and −1 °C/min for annealing. The annealed oligos were stored at −20 °C.

Tn5 enzyme was produced with minor modifications[60]. The pTXB1-Tn5 expression vector was transformed into C3013 cells (NEB) following the manufacturer's protocol. Single colonies of transformed C3013 cells were cultured 37 °C overnight (12–14 h) and seeded in 1 L LB. The culture was incubated at 37 °C for 3 h until A600 = 1.0. IPTG was added to a final concentration of 0.25 mM. Then the culture was incubated at 23 °C for 8 h to express Tn5 enzyme until A600 = 2.3 ~ 2.5. One 50 mL-bacteria pellet was resuspended by 10 mL HEGX Buffer (20 mM HEPES-KOH at pH 7.2, 0.8 M NaCl, 1 mM EDTA, 10% glycerol, 0.2% Triton X-100 with protease Inhibitors (W/O EDTA) (to 1×)). Resuspended bacteria were sonicated on a Branson sonicator: 6 cycles (25 bursts/cycle) 50% duty cycle Output=4. After sonication, bacteria were centrifuged at 13,523 × $g$, 30 min, 4 °C. The supernatant was collected and 100 µL 10% neutralized PEI (PH 7.0) was added dropwise and then centrifuged at 13,523 × $g$, 10 min, 4 °C. Next, a Chitin column was prepared and washed four times with HEGX. The supernatant from the previous step was loaded on to the Chitin column and rotated at 4 °C for 1.2 h. After incubation, the bottom cap was opened and the liquid flow-through discarded, followed by four times wash with HEGX. The annealed oligos were then diluted into HEGX buffer and loaded onto the column and incubated at 37 °C for 1 h followed by 4 °C for 2 h. The liquid flow-through was discarded followed by 4 times wash with HEGX. After the final wash, 3 mL HEGX buffer supplied by 100 mM DTT was added to the chitin and incubated at 4 °C for 48–72 h. The flow-through was collected and concentrated by Amicon Ultra centrifugal filters (MWCO 30 kDa). The concentrated protein was washed once by dialysis buffer (100 mM HEPES-KOH pH 7.2, 0.2 M NaCl, 0.2 mM EDTA, 2 mM DTT 0.2% Triton X-100, 20%Glycerol). The concentrated transposome is ready for the snATAC-Seq.

**snATAC-seq.** Combinatorial single nuclei ATAC-seq was performed as previously described with minor modifications[20]. Islets were centrifuged at 180xg for 3 min at room temperature. Supernatant was discarded. 3 ml pre-chilled lysis buffer (10 mM Tris-HCl ph7.4, 10 mM NaCl, 3 mM MgCl₂, 0.1% NP40, 0.1% Tween-20, 0.01% Digitonin) was added and the sample was homogenized by douncer on ice, followed by incubation on ice for 10 min. Islet nuclei were then filtered by a 40 um cell strainer and centrifuged 450 × $g$ for 5 min at 4 °C. The supernatant was discarded and washed once by wash buffer (10 mM Tris-HCl ph7.4, 10 mM NaCl, 3 mM MgCl₂, 0.1% Tween-20), followed by 450 × $g$ for 5 min at 4 °C. The supernatant was discarded, and nuclei were resuspended into TD buffer (10 mM Tris-HCl, 5 mM MgCl2, 10% Dimethyl formamide, 70uM Pstop) and counted by hemocytometer. Concentration was adjusted to 450 nuclei/µl. 4000 nuclei (9 µl) were dispensed into each well of a 96-well plate. 1ul unique combination of barcoded Tn5 was added to each well, mixed thoroughly and incubated at 55 °C for 15 min. To stop the reaction, 10 µL of 40 mM EDTA were added to each well and the plate was incubated at 37 °C for 15 min. Nuclei from all

wells were then combined for sorting. Twenty-five nuclei were sorted into each well of a 96-well plate with 6ul collection buffer (10 mM Tris, 10 mM NaCl, 0.02% SDS, 2% BSA). After sorting, the collection plate was incubated 55 °C for 15 min and quenched by triton-X100 to final concentration 1%. The quenched plate can be stored at −20 °C for weeks or moved to the next step for PCR amplification. For PCR, 0.6 µl primer N7** and 0.6 µl primer N5** (listed in Supplementary Table 3) were added in each well and 6 µl NEBNext High-Fidelity 2× PCR Master Mix (NEB) was added in each well. The plate was PCR amplified by 11–15 cycles (72 °C 5 min, 98 °C 30 s, (98 °C 10 s, 63 °C 30 s, 72 °C 60 s) *cycles. After PCR amplification, all wells were combined and purified with Zymo DNA Clean & Concentrator-100, followed by SPRI 1× purification. The snATAC library was sequenced on NovaSeq-6000 SP using customized sequencing primers:

snATAC.S.read1: TCGAGGACGGCAGATGTGTATAAGAGACAG;
snATAC.S.read2:GTCTCCGCCTCAGATGTGTATAAGAGACAG;
snATAC.S.index1: CTGTCTCTTATACACATCTGAGGCGGAGAC).
The sequencing read length is as follows, R1:33nt, i7:29nt, i5:28nt, R2:33nt.

**eHi-C.** We generally followed our previous easy Hi-C protocol to generate the eHi-C libraries for human α and β cells[27]. In order to make the protocol compatible with low input sorted primary cells, we made a few changes and modifications. For each library, ~20–30k sorted primary α or β cells were subjected to eHi-C processing and library construction. Briefly, the nuclei were first extracted by lysing the cells with cell lysis buffer (10 mM Tris-HCl pH 7.5, 0.2% NP-40, 10 mM NaCl, 1× proteinase inhibitor cocktail). The extracted nuclei were then subjected to HindIII digestion followed by proximity ligation with T4 DNA ligase to create spatial ligates. After proximity ligation, the DNA was extracted and prepared for the secondary restriction enzyme (DpnII) digestion. All DNA obtained was subjected to DpnII cutting before being self-ligated to form self-circles. The unligated DNA was then removed using Lambda exonuclease. To create the eHi-C library, the self-circles were re-linearized by digesting again with HindIII. All DNA recovered was applied to generate sequence-able libraries using the Illumina Truseq platform. Briefly, fragment ends were first repaired with the DNA end-repair enzyme cocktail. Then, a base "A" was added to each end by Klenow fragment (NEB, M0212) before being ligated to a modified Truseq adapter using Quick ligase (NEB, M0202). We used the regular i7 indexes as sample indexes and i5 as fragment indexes by changing it to random indexes (i5: NNNNNN).

**Cut and run.** 500–600k EndoC-βH3 cells per sample were digested into single-cell suspension with 0.05% Trypsin, EDTA 0.5 mM. We used the Cutana ChIC/CUT&RUN Kit from Epicypher (#14-1048) following manufacturer's instructions. Briefly, ConA Beads were resuspended and 11 µl of beads per reaction was transferred to a 1.5 ml tube. Beads were resuspended in 100 µl bead activation buffer per reaction and washed 1×. Then resuspended in 11 µl buffer per reaction and aliquoted 10 µl into 8-strip PCR tubes. Harvested cells were spun down at 600 × $g$ for 3 min. Cells were then resuspended in 105 µl/reaction of wash buffer and aliquoted into 8 strip tubes with the 10 µl of activated ConA beads. After 10 min. incubation, 50 µl of antibody buffer was added. HNF1A antibody for Cut&Run (Abcam #ab204306) was added at 1 µg per reaction. Tubes were incubated on nutator overnight at 4 °C.

The next day, tubes were collected and washed 2× with 200 µl cell permeabilization buffer. Beads were resuspended in 50 µl cell permeabilization buffer and 2.5 µl pAG-MNase was added. After 10 min. incubation at room temperature, beads were washed with another 200 µl cell permeabilization buffer 2×. Finally beads were resuspended in 50µl cell permeabilization buffer. 1 µl of 100 mM CaCl₂ was added to activate the enzyme. Tubes were then incubated for 2 h on nutator at 4 °C. Finally, the reaction was stopped with 33 µl of stop master mix. DNA was then purified and quantified using a Qubit fluorometer.

For library preparation, we used the CUTANA CUT&RUN Library Prep Kit from Epicypher (#12-1002) with slight modifications. During the End Repair step, we used the following modified PCR protocol to better retain small TF fragments: 20 °C for 20 min, 55 °C for 1 h, 4 °C hold. Adapter ligation and U-excision were performed following manufacturer's instructions. We used 1.75× SPRIselect beads for DNA cleanup to better retain >100 bp fragments. After indexing PCR, we then used 1.2× SPRIselect beads for DNA cleanup prior to sequencing: 150PE, 16 M reads per sample.

## Bioinformatic analysis

**Drop-Seq reads processing.** We performed raw reads processing following the instructions described in the original Drop-Seq publication[19]. Briefly, the sequenced Drop-Seq libraries yield 150-base paired-end reads (PE150). The first 20 bp of read 1 are cellular and molecular barcodes (base 1–12 cell barcode, base 13–20 UMI). We trimmed base 21–150 of read 1 before further analysis. We first removed all data with the quality score of read 1 (base 1–20) lower than 10. Read 2 was trimmed at the 3′ end to remove poly A tails of at least 6 bases and trimmed at the 5′ if template switching oligo (TSO) adapter sequences appear. Clean reads were then aligned to hg19 using STAR v2.5.1 with default settings. We only kept uniquely mapped reads on gene exons. We next filtered out PCR duplicates with the same coordinates, cell barcode, and UMI. We then grouped the reads by cell barcode and generated the digital UMI-count matrix after counting transcripts for every gene with every cell barcode (Supplementary Data 5).

**snATAC-seq reads processing.** The paired-end raw reads were aligned to hg19 using bowtie2 v2.2.6 with default settings. We next filtered out mitochondrial reads and PCR duplicates with the same coordinates or cell barcode using picard v1.93 and converted bam files to bedgraph files using bamtobed v1.2.0. MACS2 v2.2.7.1 was used to call peaks with default parameters with the ratio of reads on ATAC peak over 0.15 and the total reads over 1500 as the cutoff to define a qualified cell (Supplementary Data 5).

**scRNA-seq and snATAC-Seq clustering and QC analysis.** Here, we made use of both the single-cell RNA and ATAC data to achieve the best clustering resolution and consistency. We have previously comprehensively profiled the human single-cell transcriptome for nine donors which clearly resolved the islet cell types in transcriptome space[10]. It is known that snATAC data are sparser than scRNA-seq but show substantial correlations with the gene expression. Therefore, in this study, we used scRNA-seq data for the initial clustering and annotation, and then applied a canonical correlation analysis (CCA)-based co-embedding to propagate the information into the snATAC-seq data.

The scRNA-seq filtering and clustering strategy for islets is as reported earlier[10] with a hormone-based doublet filtering algorithm. Briefly, we first filtered out STAMPs (single cell transcriptome attached to microparticles) expressing two hormones (Supplementary Fig. 1a) before clustering analysis. In this step, one STAMP is considered as a doublet if it has two hormone genes that are highly expressed (defined as more than 10% of the median expression level in the positive population). As discussed previously, this step is important as the percentage of doublets in the primary islet is significantly greater than estimated from species mix experiments due to the nature of tissue adhesion. We used *Seurat v3* package for clustering analysis with default parameters. In *Seurat*, *PCA* was performed with the 500 informative genes. Using PC1 to PC10, cells were embedded in a K-nearest neighbor (*KNN*) graph. Smart local moving algorithm (*SLM*) was applied to group cells into communities. PC1 to PC10 were used as input to visualize cell clusters in two-dimensional *UMAP* space. We used *Seurat FindMarker* function to find marker genes of each cell cluster, and defined cell types based on previous datasets and literature. After clustering, we performed a secondary filtering by

removing cells with hormone genes inconsistent with their cell type classification (>15 transcripts) (Supplementary Fig. 1b). We aggregated the raw count matrix into pseudo-bulk *RPKM* matrix (Cell by cell types) for both datasets and performed the correlation analysis (Supplementary Fig. 1c).

We used the snATAC (Cell-GeneScore) count matrix and the filtered scRNA-seq count matrix as input for co-embedding clustering. Both snATAC and scRNA count matrices are normalized and combined into a common canonical correlation analysis (CCA)-based space using *RunCCA* in *Seurat v3*, followed by *L2* normalization. We then used half of the RNA data on the common *CCA* space which has known cell-type annotations from the previous step, to train a Support Vector Machine (*SVM*) model with radial kernel and cost=10. We tested the remaining half of the RNA data as validation which yielded up to 99% assignment accuracy. We then applied the trained *SVM* model to assign the cell type labels for the snATAC data on the same *CCA* space. To filter out potential doublets in snATAC data, we further reconstructed a k-nearest RNA neighbor graph for each snATAC. The Euclidian distance on *CCA* space was used to weight the *KNN* graph. We computed a unambiguity score that is defined as the average distance ratio between the first nearest cell type versus second nearest cell type ($k = 20$). We took the unambiguity score 10 as a cut off that filtered out the potential doublets from *SVM*-annotated snATAC data (Supplementary Fig. 1g). After snATAC data are assigned and filtered, we reconstructed ATAC fragment-level data (bam file) and call peaks separately in a cell-type specific manner (using *MACS2*). The union of all peaks called from each cell type followed by filtering out blacklist regions are used to produce a new Cell-Peak matrix for downstream analysis and visualization.

**HNF1A CUT&RUN reads processing.** We used the default settings of *bowtie2 v2.2.6* to align the paired-end raw reads. Then we use *samtools v1.3.1* to sort and extract uniquely mapped raw reads. Then we filtered out PCR duplicates with the default parameter by using *picard v1.93*, and *MACS2 v2.2.7.1* was used to call peaks. After having the non-duplicated sorted bam file we converted bam files to bedgraph files using *bedtools v 2.25 (bamtobed,genomecov)*. Finally we use *bedGraphToBigWig* to convert bedGraph to a bigWig file and upload to UCSC for track graphing.

## eHi-C data pre-processing for QC and performance analysis

**eHi-C data pre-processing.** The sequencing data were mapped to human reference genome hg19 using Bowtie v1.1.2. Because nearly all the mappable reads start with HindIII sequence AGCTT, we trimmed the first 5 bases from every read, took the next 36 bases, and added the 6-base sequence AAGCTT to the 5′ of every read before mapping using the whole 42 bases. After mapping, we further filtered the reads requiring the positions of both ends to be exactly at the HindIII cutting sites. After removing PCR duplications, we next split all the remaining reads into three classes based on their strand orientations (same-strand, inward, or outward). and estimated the total number of real cis-contacts as twice the number of valid same-strand pairs.

**Compartments calling.** We performed compartment level analysis following the original method[61]. We divided the genome into 250 kb bins and generated the contact matrices between bins for each chromosome and next normalized the matrix by genome distance. Briefly we calculated the average of all interaction values with the same distance.

We next generated the Pearson's correlation matrix from the distance-normalized matrix. And performed the principal component analysis on the correlation matrix then assigned the genome into two compartments depending on whether the PC1 of a bin is a negative or positive value. The TSS data were used to determine compartment A and B (More TSS sites: compartment A; fewer TSS sites: compartment B).

**Loop calling.** We applied *HiCorr*[27] to remove bias from eHi-C fragment pairs data and got contacts at ~5 kb anchor level, and then used *DeepLoop*[28] to enhance contact signals. Then we ranked the contacts by their interacting strength value output from *DeepLoop* and took the top 300 K pixels as the strong contact signals (Supplementary Data 6).

**Specific gene expression with loops.** For the genes specifically expressed in α cells, we further annotated the source of specificity by checking: **a.** If the gene located in α specific A compartment. **b.** If there are α specific ATAC peaks on the TSS of the gene. **c.** If the TSS of genes linked to α specific ATAC peaks by α loops. **d.** If the TSS of the gene is linked to ATAC peaks by α specific loops. **e.** If the TSS of the gene is linked to α specific ATAC peaks by α specific loops. We performed the same analysis for β specifically expressed genes.

**Downstream analysis**

**snATAC peak and motif analysis.** We started with the snATAC count matrix (all cells versus union peaks identified above). We firstly aggregated single cell counts in the same cell type and computed *RPKM* for each peak in each cell type. Next, we calculated the maximum peak in endocrine (β, α, δ, PP), as well as the maximum peak in non-endocrine cells (Acinar, PSC, Duct). For any given peak, the *RPKM* ratio between maximum in one of the endocrine and non-endocrine cell types is computed. We classified the peaks with this ratio >2 as endocrine-specific peaks, whereas the ones <0.5 as non-endocrine specific peaks. The rest of the peaks were classified as common peaks. Both the endocrine and non-endocrine specific peaks were further grouped by *K-means* clustering.

We performed two levels of motif analysis. One was at cell level, to study the overall motif enrichment in a group of cells. The other was at peak level, to resolve the enriched motif in a group of peaks. For cell level analysis, we applied chromVar that computed the z-score of motif enrichment in each single cell given the cell-peak count matrix. The peak level enrichment followed the identification of endocrine and non-endocrine peak clusters (C1 ~ C13). First, we generated the background motif frequency by scanning over all sequences of all peaks for human Hocomoco v11 motif database (FIMO, MEME suit). For each group of peaks, we computed the observed motif occurrence by FIMO scanning. *P*-value < 1e-6 was used as the cutoff to define a significant motif. By comparing the occurrence of the observed motif count to the background, we computed the motif enrichment score by *Binomial test*. The *p*-value is further adjusted into q-values.

**Peak-gene analysis.** We performed three ways of peak-gene assignment. One is distance based. All peaks are annotated using *Homer* (*annotatePeaks.pl*) that assigns a gene to the given peak by the distance from the TSS to peak (nearest in principle). The other is Hi-C loop based, which is specified in the following section. These two measurements are both further applied in downstream regulatory network analysis. The third method is correlation based. We computed the *Pearson's correlation* between the aggregated peak *RPKM* versus expression *RPKM* across seven different cell types. We chose a 1 Mb window for the potential peak-gene pair candidates. And only classified the ones with Pearson's correlation >0.7 as potential regulatory peak-gene link (Supplementary Fig. 2).

**RNA-RePACT.** We have developed RePACT (<u>Re</u>gressing <u>P</u>rincipal components for the <u>A</u>ssembly of <u>C</u>ontinuous <u>T</u>rajectory) as a general method to sensitively identify disease relevant gene signatures using single cell RNA-sequencing data[2]. The key step is to find the best trajectory to rank single cells (e.g., β cells) reflecting the change of disease status.

We took β cells from all donors, performed PCA analysis, and picked the top 10 PCs. We next performed a logistic regression for binary variables such as (T2D/Healthy).

With the regressed β values, we computed the T2D-index for every cell. The T2D-index is used to rank the cells, and its value indicates how far a cell is transformed toward T2D status. To identify genes associated with the T2D trajectory. We grouped all cells into 20 bins with equal T2D-index intervals; every bin contains hundreds of single cells. For every gene, we then calculated the average transcript counts from cells in each bin and obtained a vector of 20 values. A linear regression was performed between the average transcript counts and the index values of the bins with *p*-value. The *p*-values of all genes were adjusted with Bioconductor package *q* value to obtain *q*-values. Genes with *q*-value <0.05 were called significant trajectory genes.

**ATAC-RePACT.** The ATAC-RePACT follows the same principle as RNA-RePACT and is generalized into the epigenetic level. This algorithm aims to sensitively identify disease relevant epigenetic signatures using snATAC data. In general, the first step is computationally isolating one cell type of interest. If multiple cell types are of interest, the analysis is recommended to be done separately to minimize confounding effects from the cell-type level difference. In the current study, we first used the β cell from snATAC data (starting with a cell-peak matrix as discussed above). We performed the TF-IDF normalization and dimension reduction into the LSI space (Latent Semantic indexing space, dims = 50). On the LSI space, we built a logistic regression for binary phenotypes (T2D/Healthy). The regression model reconstructed the disease/phenotype oriented single cell trajectory along which the cells are gradually changing cell states (chromatin accessibility in this case). Along this trajectory, we binned all cells into 20 bins. We used another linear regression to model the chromatin accessibility changing trend associated with the phenotype. We defined the peaks with *q*-value 0.01 and slope >0.5 as gain of accessibility, whereas the peaks with slope less than −0.5 as loss of chromatin accessibility.

We performed both cell level and peak level transcription factor motif enrichment analysis. Briefly, we used Chromvar to compute the motif enrichment Z-score in each single cell and applied a linear model to measure the motif usage change along the trajectory. Also, gain of accessibility and loss of accessibility peaks were used for peak level enrichment analysis by FIMO as described above.

**Intra-donor-level RePACT.** The General *RePACT* is designed to take multiple donors that have different phenotypes (i.e., T2D/Healthy; variable BMI; etc.). These phenotypes are critical as the input to train the phenotype-oriented single-cell trajectory. To further analyze the phenotype associated single cell heterogeneity within a given donor, we developed the following method for the intra-donor-level RePACT. Both scRNA and snATAC follow the same principle. We started with the dimension-reduction space of total cells including all donors that have already built the general RePACT trajectory (PCA space for RNA, LSI space for ATAC, respectively). On the same space, we iteratively focused on each donor and regrouped the single cells into 20 bins. The gene expression change (or chromatin accessibility change) across the 20 bins in the given donor is then modeled by linear regression and a *p*-value and slope were computed to describe the heterogeneity trend for each gene in each donor. Next, *Fisher's* method was used to combine the *p* values across all donors to measure the reproducibility of the heterogeneity in different donors.

$$Combined\ pvalue = P\left(X > -2 \times \sum_k \log(pvalue_k) | X \sim \chi2(df = 2 \times k)\right)$$

Where k is the number of donors, $pvalue_k$ is the individual RePACT *p* value. The combined *p*-value after Fisher's Method was further adjusted by qvalue. *qval(Combined pvalue)*. We defined the genes with Fisher's method *q*-value <0.01 as intra-donor heterogeneous RePACT genes, which suggest a reproducible intra-donor variability that is consistent with the global phenotype association. The intra-donor

heterogeneity score was defined as -$\log 10(qval(Combined\ pvalue))$ based on the chi-square distribution.

**HNF1A target gene identification.** For the T2D trajectory peaks with *HNF1A* motifs (motif calling as discussed above), we first find their nearest genes, and then check if the peaks are connected to some distal genes by chromatin loops. And we further overlap these genes with T2D trajectory genes to get the *HNF1A* target gene prediction.

**GSEA analysis.** Gene function enrichment analysis was performed by integrating MSigDB (v.5.2). All functional term lists were read into R using the package 'gage'. For any given group of genes, a binomial test was performed iteratively through all annotated functional terms. *P*-values for enrichment were further adjusted using the qvalue package. Enrichment terms were ranked by *q* value. The most representative top terms were selected and visualized using heatmaps. On each heat map, *q*-values of enriched terms were visualized by color intensity as enrichment scores.

**Reanalysis of Patch-seq in islet β cells.** The cell-gene expression matrix and cell meta data with variable electrophysiological measurements for each cell were obtained from GEO accession GSE124742. To validate if the identified target genes were regulated by *HNF1A*, we separated β cells by if *HNF1A* transcripts were detected and compared the normalized expression of the target genes between the two groups. To explore the genes that were associated with electrophysiological measurements, for each gene, we first divided β cells into two groups, cells with the gene expressed, and cells without the gene expressed. If there are more than half of β cells expressing the gene, then we ranked the cells by expression and regroup the cells to high expression and low expression. We further compared each electrophysiological measurement between these two groups of cells. All current measurements were transformed to positive values regardless of flow direction across the cell membrane. Wilcox.test was used to compute *p*-values.

### Reporting summary

Further information on research design is available in the Nature Portfolio Reporting Summary linked to this article.

## Data availability

The raw scRNA-seq, snATAC-seq, eHi-C, and Cut&Run data generated in this study are available in the NCBI GEO database under accession code GSE195523 and GSE234754. The cell type specific genes are provided in Supplementary Data 1. The cell type specific ATAC peaks are provided in Supplementary Data 2. The T2D trajectory dynamic genes and peaks, intra-donor, inter-donor genes and peaks are provided in Supplementary Data 3. The peak-gene regulatory links are provided in Supplementary Data 4. The QC summary for single cell data is provided in Supplementary Data 5. The chromatin loops identified in α and β cells data are provided in the Supplementary Data 6. Source data are provided with this paper. The web app to visualize the gene expression and chromatin accessibility across cell types, and RNA-RePACT and ATAC-RePACT is available at https://hiview.case.edu/public/BetaCellHub.

## Code availability

The code is available is available at Zenodo (https://doi.org/10.5281/zenodo.8264879) and GitHub (https://github.com/JinLabBioinfo/RePACT).

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

## Acknowledgements

This work was supported by grants from NIH R01DK113185 (to Y.L.), R01HG009658 and R01CA267872 (to F.J.), R01DK131437, R01HG012384, and UG3NS132061 (to Y.L. and F.J.); a pilot award from Clinical and Translational Science Collaborative (CTSC) at Case Western Reserve University (VSN639001 to Y.L.). F.J. is also supported by a subaward from University of Miami (NIH U01AG072579) and a Cancer Data Sciences pilot grant from Case Comprehensive Cancer Center Support Grant (NIH P30CA043703). A.G. was supported by a NIH MSTP training grant (T32GM007250) and a Functional Genomics Training Program grant (T32GM135081). We thank *Lifebanc* for providing some additional pancreases samples which are used for HNF1A antibody verification. This work made use of the High-Performance Computing Resource in the Core Facility for Advanced Research Computing at Case Western Reserve University. We also gratefully acknowledge the insightful discussions with Scavuzzo M. and Wang L.

## Author contributions

Y.L. and F.J. conceived the project. C.W., A.G. performed the experiments. L.L., L.K., Y.W., P.H., E.M., J.X., S.L. and K.L. also contributed to the experiments. C.W., A.G., and Shanshan Z. carried out the data analysis. P.G., X.L., D.P., Saixian Z., and K.Y. also contribute to the data analysis. Y.L., F.J., C.W., A.G., Shanshan Z. wrote the paper with inputs from all authors.

## Competing interests

The authors declare no competing interests.
