## [Peer Review File · Nature Communications]

Single cell multiomic analysis reveals diabetes-associated β -cell heterogeneity driven by HNF1AREVIEWER COMMENTS

Reviewer #1 (Remarks to the Author):

In this manuscript the authors perform scRNAseq (drop-seq), snATAC-seq and HiC in islet cells from human donors with and without T2D. The study is therefore very comprehensive in terms of having multiple 'omics' measurements performed in cells from the same islet preparations. Other scRNAseq/snATAC studies exist both in terms of cells and donors, however this study is unique in having both measurements in the same islet preparations, as well as chromatin conformation for a- and b-cells. The authors also investigate patch-seq data to come up with a mechanistic model by which HNF1A dysregulation might be involved in beta-cell dysfunction in T2D. They find that these cellular states exist within and across donors and become more abundant in T2D. The paper provides novel data as well as an interesting framework for data analysis to go from gene expression and genome regulation to physiological impact in the cells. The manuscript is novel and interesting enough to be published in Nature Communications. There's a few aspects that are not clear and would recommend the authors to address them.

1) The authors identify genes with either high inter-donor or high intra-donor variation. This distinction is an interesting angle of the paper, but it also brings up the possibility that some inter-donor genes are driven by donor effects and not T2D. Can the authors control for this within the RePACT model? If not, is this addressed in some other way? Are intra-donor variable genes more reliable markers?

2) The authors use a method that they previously developed (RePACT) to draw a continuous disease trajectory and pick cellular states within the cell pool of each donor. In theory this could help identify subpopulations of cells in their path towards a 'diseases' T2D cell-state. This is particularly useful when a continuous disease metric exists, however it is less obvious in a dichotomous label healthy/T2D. Have the authors used some other disease metric such as Hb1AC? Otherwise, can they clarify how this method works without having more granularity in the phenotype? If they do have HbA1C data can they superimpose it on the trajectory?

3) The term multiomic is slightly confusing as this name is now being popularized by a 10X kit where scRNAseq+snATACseq is performed in the same cell. The authors are quite clear on their methodology, but I would still expect that some people might have the wrong expectations when picking up the paper. I don't have a specific request in relation to this, as 'multiomics' is a rather generic term, but the authors/editors might want to be aware of this and maybe adjust the writing in some parts.

4) Some claims on "first-time novelty" and "strength of a model" are probably not needed (page 10 lines 286 and lines 294). The novelties of the paper should already be clear to the readers from the data presented.

Joan Camunas-Soler

Reviewer #2 (Remarks to the Author):

In this manuscript, the authors performed scRNA-seq and scATAC-seq on healthy and T2D human islets, as well as HiC on α and β cells. The authors assumed a prerequisite that fluctuations in the proportion of heterogeneous subpopulations are related to T2D. To identify genes that are differentially expressed in healthy β cells and also differentially expressed in T2D patients (intra-donor heterogeneity), the authors designed an algorithm to force β cells into T2D-like and T2D-unlike subpopulations and applied the algorithm to scRNA-seq and scATAC-seq datasets. Then, the authors identified HNF1A as a T2D-related gene according to its CREs and binding motifs. In addition, the authors suggested that HNF1A drives the intra-donor heterogeneity of β cells.

The authors' idea of identifying differentially expressed genes in T2D from highly variable gene sets is interesting, but further proof or discussion is required to determine whether these gene sets reflect the heterogeneity of β cells. The flaw in this work may lie in the setting of the authors' prerequisites. A heterogeneously expressed single gene or gene module may not be able to effectively distinguish β cell subpopulations. To conclude that β cells can be subdivided into HNF1A-high and HNF1A-low subpopulations, the authors should confirm that HNF1A and its downstream gene module were consistently differentially

expressed in these subpopulations and that these modules are associated with islet structure or specific functions. HNF1A may be just a highly variable gene in β cells, and β cells are dynamically transition between HNF1A-high and HNF1A-low expression states.

Major points:

1. The authors analyzed the differences in chromatin structure between α and β cells, but this does not appear pertinent to the main topic. In addition, the authors should elaborate on why chromatin compartments of α and β cells are significantly distinct, despite the absence of cell-specific transcripts or open chromatin in the variable compartments.
2. Since the healthy donors also exhibited T2D-related intra-donor heterogeneity, it is preferable to use immunofluorescence to validate T2D-related heterogeneity in both healthy and T2D samples.
3. In Fig. 4c, it appears that the T2D pseudotime index fluctuates considerably among healthy donors. Is there a correlation between the T2D index and the risk of T2D in these healthy donors?
4. Please show the single cell expression levels of HNF1A in different β -cell populations in healthy and T2D samples.
5. Although the authors predicted the target genes of HNF1A, they did not perform HNF1A ChIP-seq to verify the target genes.
6. Can the authors apply RePACT on HiC dataset to find differential loops? Do the loop between HNF1A and its CRE change following “intra-donor heterogenous”?

Minor points:

1. Line 129 cited Fig. 3b. Should be Fig. 3d.
2. Starting from line 228, HNF-1 α should be changed to HNF1A, consistent with the previous text.

Reviewer #3 (Remarks to the Author):

Weng et al. present a comprehensive multi-omic study of islets from healthy and T2D donors, including scRNA-seq, snATAC-seq, and Hi-C. The authors extend their previous RePACT method to snATAC-seq and show beta cell heterogeneity at the epigenomic level, in addition to the transcriptomic level. Importantly, the authors identify HNF1A as a putative driver for beta cell heterogeneity through the integration of multi-omic datasets and infer its function through the HNF1A-FXYD2 pathway. Overall, the study is well done, and the figures are of high quality. However, there are a few concerns that need to be addressed.

1. As far as I noticed, basic QC metrics such as the number of reads and the number of genes/peaks per cell for scRNA-seq and snATAC-seq data are not provided. Including these statistics would provide a better understanding of the generated dataset.
2. While the discrepancy between cell-type-specific ATAC peaks and compartment switches is intriguing, the 3D genome part of the manuscript seems somewhat unrelated to the overall story, except for the loops being used to link ATAC-seq peaks to genes. Additionally, the list of loops should be provided as a supplementary table, as it can be a valuable resource for the field.
3. For Figure 4d, while the heatmap suggests that this study has similar gene patterns with Fang et al., there is no quantitative measure to show the consistency.
4. For the peak-to-gene assignment in Figure 6, it is unclear whether the peak-to-gene relationship is always one-to-one. If there are multi-to-multi assignments, how they add up for the T2D trajectory is unclear.
5. While it is nice to have the RePACT code provided, there are no codes to replicate major results of the paper. It is critical to have these codes for reproducibility.

We thank all the reviewers for their time and insightful comments. Below are our responses to their comments.

Reviewer #1 (Remarks to the Author):

In this manuscript the authors perform scRNAseq (drop-seq), snATAC-seq and HiC in islet cells from human donors with and without T2D. The study is therefore very comprehensive in terms of having multiple 'omics' measurements performed in cells from the same islet preparations. Other scRNAseq/snATAC studies exist both in terms of cells and donors, however this study is unique in having both measurements in the same islet preparations, as well as chromatin conformation for a- and b-cells. The authors also investigate patch-seq data to come up with a mechanistic model by which HNF1A dysregulation might be involved in beta-cell dysfunction in T2D. They find that these cellular states exist within and across donors and become more abundant in T2D. The paper provides novel data as well as an interesting framework for data analysis to go from gene expression and genome regulation to physiological impact in the cells. The manuscript is novel and interesting enough to be published in Nature Communications. There's a few aspects that are not clear and would recommend the authors to address them.

We thank the reviewer for the supportive comments.

1) The authors identify genes with either high inter-donor or high intra-donor variation. This distinction is an interesting angle of the paper, but it also brings up the possibility that some inter-donor genes are driven by donor effects and not T2D. Can the authors control for this within the RePACT model? If not, is this addressed in some other way? Are intra-donor variable genes more reliable markers?

We thank the reviewer for bringing up this important question. In fact, "donor effects" will impact both "inter-donor" and "intra-donor" marker genes because both types of marker genes are identified from the same T2D trajectory. But RePACT alleviates "donor effects" for both types of marker genes.

The reviewer is worried that "donor effect", which refers to T2D-irrelevant inter-donor variation, may affect the identification of inter-donor T2D signature genes. We believe RePACT should alleviate "donor effects" because in RePACT, we establish a T2D trajectory using β -cells from all individuals: both intra-donor variation and inter-donor variation contribute to the T2D trajectory. This is why RePACT leads to a more stable trajectory identification, especially when only a limited number of donors are available. We have discussed this point in our original RePACT paper (Fang et al, Cell Reports, 2019 PMID:30865899). In other words, a robust trajectory is key to identify disease marker genes, regardless of if they are "intra-donor" or "inter-donor". Since RePACT improves the robustness of the T2D trajectory, it improves the accuracy of both types of marker genes.

On the other hand, “donor effects” still inevitably affect the T2D trajectory. Like any correlation analysis, the best way to control for donor effects is still to obtain more samples to control all possible confounding factors, including age, gender, BMI, ethnic group, medical history, *etc.* In practice, donor availability is always a limitation.

One approach we already took to address the “donor effects” issue is to check if the T2D marker genes from two independent cohorts agree with each other. In **Figure 4d**, we compared the results from this study and Fang *et al.* and found consistent trends even though both studies only involve a relatively smaller number of donors.

The reviewer raised an interesting question if “intra-donor” variable genes are more reliable. We have included a new analysis examining the overlap of “intra-donor” and “inter-donor” genes to the genes called from Fang *et al.* (**Reviewer Fig. 1** below, we have included this figure as the new **Fig. 4e** in the revision). We found that both “intra-donor” and “inter-donor” genes are significantly enriched with the marker genes called by Fang *et al.* Furthermore, the “inter-donor” variable genes are enriched with mitochondrial and translation functions (**Fig. 4k**), which are known diabetes relevant functional categories. Therefore, we do not think the “inter-donor” genes are less reliable.

Reviewer Fig. 1. Reproducibility between this study and Fang et al.

2) The authors use a method that they previously developed (RePACT) to draw a continuous disease trajectory and pick cellular states within the cell pool of each donor. In theory this could help identify subpopulations of cells in their path towards a ‘diseases’ T2D cell-state. This is particularly useful when a continuous disease metric exists, however it is less obvious in a dichotomous label healthy/T2D. Have the authors used some other disease metric such as Hb1AC? Otherwise, can they clarify how this method works without having more granularity in the phenotype? If they do have HbA1C data can they superimpose it on the trajectory?

RePACT uses a classic general linear model (GLM) which supports both continuous and dichotomous dependent variables (Y). Specifically, in GLM we use linear regression to

model continuous variables such as HbA1C and use logistic regression to model binary variables such as healthy/T2D.

To address the reviewer’s question, we applied RePACT using HbA1C as the dependent variable. HbA1C is a commonly used measurement for T2D diagnosis; our T2D donors indeed have higher HbA1C. We observed the “shift” among cells from high-HbA1C and low-HbA1C donors (**Reviewer Fig. 2a** below). We then built the HbA1C trajectory and called genes that are up- or down-regulated. As expected, the marker genes from T2D and HbA1C trajectories agree very well; there are no contradictions between the T2D and HbA1C RePACT analyses (**Reviewer Fig. 2b-c**, below).

Reviewer Fig. 2. RePACT analysis using HbA1C as dependent variable. (a) After building a β -cell HbA1C trajectory with RePACT, the violin plot compares the distribution of HbA1C pseudo-index for β -cells from each donor. (b) Left heatmap: the expression changes of the top up/down regulated genes along the HbA1C trajectory. Right heatmap: the expression changes of the same genes along the T2D trajectory. (c) Venn diagram showing the overlap of genes that are independently identified in HbA1C trajectory and T2D trajectory.

3) The term multiomic is slightly confusing as this name is now being popularized by a 10X kit where scRNAseq + snATACseq is performed in the same cell. The authors are quite clear on their methodology, but I would still expect that some people might have the wrong expectations when picking up the paper. I don’t have a specific request in relation to this, as ‘multiomics’ is a rather generic term, but the authors/editors might want to be aware of this and maybe adjust the writing in some parts.

We thank the reviewer for the reminder and have limited using the word “multiomic” in the manuscript.

4) Some claims on “first-time novelty” and “strength of a model” are probably not needed (page 10 lines 286 and lines 294). The novelties of the paper should already be clear to the readers from the data presented.

We thank the reviewer for the reminder and have limited using these words in the manuscript.

Joan Camunas-Soler

Reviewer #2 (Remarks to the Author):

In this manuscript, the authors performed scRNA-seq and scATAC-seq on healthy and T2D human islets, as well as HiC on α and β cells. The authors assumed a prerequisite that fluctuations in the proportion of heterogeneous subpopulations are related to T2D. To identify genes that are differentially expressed in healthy β cells and also differentially expressed in T2D patients (intra-donor heterogeneity), the authors designed an algorithm to force β cells into T2D-like and T2D-unlike subpopulations and applied the algorithm to scRNA-seq and scATAC-seq datasets. Then, the authors identified HNF1A as a T2D-related gene according to its CREs and binding motifs. In addition, the authors suggested that HNF1A drives the intra-donor heterogeneity of β cells.

The authors' idea of identifying differentially expressed genes in T2D from highly variable gene sets is interesting, but further proof or discussion is required to determine whether these gene sets reflect the heterogeneity of β cells. The flaw in this work may lie in the setting of the authors' prerequisites. A heterogeneously expressed single gene or gene module may not be able to effectively distinguish β cell subpopulations.

To conclude that β cells can be subdivided into HNF1A-high and HNF1A-low subpopulations, the authors should confirm that HNF1A and its downstream gene module were consistently differentially expressed in these subpopulations and that these modules are associated with islet structure or specific functions. HNF1A may be just a highly variable gene in β cells, and β cells are dynamically transition between HNF1A-high and HNF1A-low expression states.

We want to clarify that we do NOT intend to claim a discrete β -cell subpopulation. Reflecting this point, our model in **Fig 7k** shows a continuous HNF1A expression variation.

We agree with the reviewer that HNF1A may be a highly variable gene in β -cells. It remains to be determined whether the continuous heterogeneity of HNF1A is a stable condition or represents dynamic cells transitioning between HNF1A-low or HNF1A-high states. We also agree with the reviewer that one heterogeneously expressed single gene or gene module cannot define a β -cell subpopulation. In our opinion, "subpopulations" shall be stable cell states and from single cell data, the cells need to form discrete clusters and every cluster should contain cells from multiple individuals. In our original RePACT paper (Fang et al,

PMID: 30865899), we already discussed about the lack of major β -cell subpopulations, and in this study, we did not observe such β -cell clusters either.

On the other hand, we believe: (i) β -cell heterogeneity can be either continuous or discrete; (ii) but continuous heterogeneity can still cause molecular consequences with disease relevance; (iii) discrete subpopulations are not necessarily disease-relevant even if they exist. Taken together, although it is always attractive to claim stable “subpopulation”, it is not a necessary concept for the understanding of disease biology. A major novelty of our study is that we explicitly investigated the continuous β -cell heterogeneity within the same donor and identified HNF1A as a disease-relevant heterogeneity driver. Our RePACT method is designed to dissect the molecular signature of continuous cellular heterogeneity; HNF1A and its target genes are all discovered by RePACT. Our flow cytometry data also showed that the protein level of HNF1A is continuous; there is no clear distinction between HNF1A-low or HNF1A-high subpopulations (**Fig. 7a, Supp Fig. 5c**).

Importantly, we provide evidence that this continuous HNF1A heterogeneity is indeed functional. We verified a number of HNF1A target genes with consistent continuous transcriptional or epigenetic heterogeneity (**Fig. 6a,c,d, Supp. Fig. 5a-b**). For some of the target genes (if antibody is available), we also used flow cytometry to show that they have consistent intra-donor heterogeneity at protein levels (with high levels of target genes in the cells with high HNF1A level, **Fig. 7c,d, Supp. Fig. 5d,e**). Regarding the function of HNF1A-driven heterogeneity, we leveraged the PATCH-seq data and found that the expression of *HNF1A* (and nearly all its target genes) is associated with low sodium current, likely through upregulating *FXRD2* gene (**Fig. 7e-k, Supp Fig. 7b**). We also showed that knocking down *HNF1A* in EndoC cells leads to less insulin secretion (**Fig 7b**).

Taken together, we have established a continuous HNF1A-driven β -cell heterogeneity. It does cause consistent changes of target gene expression with functional consequences. We have added more discussion in the revised manuscript to clarify this conclusion.

Major points:

1. The authors analyzed the differences in chromatin structure between α and β cells, but this does not appear pertinent to the main topic. In addition, the authors should elaborate on why chromatin compartments of α and β cells are significantly distinct, despite the absence of cell-specific transcripts or open chromatin in the variable compartments.

In this revision, we moved some of the Hi-C figure panels to supplementary figures and shortened some text to improve the coherence of the paper.

We performed Hi-C as a part of the multiomic dataset to connect epigenome (ATAC peaks) to transcriptome. The comparison between α - and β -cells proves the association between chromatin loops, enhancers, and gene expression.

The reviewer might have some misunderstanding because we do show consistent differences between cell type specific compartments and transcription and open chromatin. Specifically, after calling α - or β -cell-specific compartment A in **Supp. Fig. 3c**, we observed that β -cell-specific compartments have more open chromatin and higher gene expression in β -cell (**Supp. Fig. 3d**, left two panels), and α -cell-specific compartments also have more open chromatin and higher gene expression in α -cell (**Supp. Fig. 3d**, right two panels).

The reviewer might be referring to the observation that a majority of α - or β -cell-specific ATAC peaks are located within regions without compartment differences (**Supp. Fig. 3e**). This is not too surprising because the activation of one or a few enhancers or promoters is probably not adequate to cause major alteration of high-order genome architecture at compartment level; more substantial changes of genome activity, such as the activation of multiple enhancers or promoters, are most likely necessary.

2. Since the healthy donors also exhibited T2D-related intra-donor heterogeneity, it is preferable to use immunofluorescence to validate T2D-related heterogeneity in both healthy and T2D samples.

Because the HNF1A-driven β -cell heterogeneity is continuous, immunofluorescence will not be able to clearly distinguish the quantitative differences between β -cells. This is why we use flow cytometry to measure the quantitative intra-donor heterogeneity. The **Fig. 7a** and the new **Supp Fig. 5c** address the question about T2D related heterogeneity. In these figures, we compared the distribution of HNF1A signal in the β -cells from 8 different donors and found that β -cells from T2D samples tends to have lower HNF1A signal. It should be noted that as we discussed above, the HNF1A signal in β -cells are mostly continuous.

3. In Fig. 4c, it appears that the T2D pseudotime index fluctuates considerably among healthy donors. Is there a correlation between the T2D index and the risk of T2D in these healthy donors?

This is an interesting question. Because the islet samples are from deceased donors, we cannot obtain definitive information if the healthy donors have T2D risk.

HbA1C and BMI are two measurements potentially relevant to T2D risks in healthy donors. In our study, the healthy donors only have a narrow range of HbA1C level 5.2~5.8% (**Fig 1b**), and we did not find any relationship between HbA1C and T2D index among healthy people (data not shown). On the other hand, there is a trend that the healthy BMI-high donors have higher T2D index (**Reviewer Fig. 3a**). We also performed RePACT among healthy donors using BMI and identified BMI trajectory genes; there is a moderate overlap between BMI and T2D trajectory genes (**Reviewer Fig. 3b**). These observations are consistent with the idea that BMI is a risk factor for T2D among healthy people. However,

these results are still preliminary, and it remains unclear if other factors may cause the fluctuation of T2D index among healthy donors.

Reviewer Fig. 3. RePACT analysis using BMI as dependent variable. (a) The violin plot compares the distribution of T2D pseudo-index for β -cells from 7 healthy donors. The donors are ranked by their BMI values. **(b)** Left heatmap: the expression changes of the up/down regulated genes along the BMI trajectory. Right heatmap: the expression changes of the same genes along the T2D trajectory. **(c)** Venn diagram showing the overlap of the genes that are independently identified in BMI trajectory and T2D trajectory.

4. Please show the single cell expression levels of *HNF1A* in different β -cell populations in healthy and T2D samples.

As discussed above, we did not observe discrete β -cell subpopulations and believe *HNF1A* represents a continuous cellular heterogeneity.

In our scRNA-seq data, the number of detected *HNF1A* transcripts is very low. In fact, even for Patch-seq, which uses a high-coverage one-cell-per-tube protocol, the dropout rate of *HNF1A* is still high: only 33 out of the 358 β -cells in Patch-seq data detected *HNF1A* transcripts. The table below shows the number of cells that can detect *HNF1A* transcript in every donor from our scRNA-seq data: 16 out of 2,824 healthy β -cells detected *HNF1A* transcripts while 5 out of 1,977 T2D β -cells detected *HNF1A* transcripts. This represents a fold change 2.24 and a marginal p-value 0.07 (binomial test).

We also use the snATAC-seq signal at *HNF1A* promoter as an indication of gene activity. 450 out of 7,668 healthy beta cells show open chromatin at the *HNF1A* promoter, while 213 out of 5,879 T2D beta cells show open chromatin at the *HNF1A* promoter (FC=1.62, p-value=1.8e-09). These results support our model that β -cells from healthy donors express more *HNF1A*.

	Donor β cell	HT1	HT2	HT3	HT4	HT5	HT6	HT7	T2D1	T2D2	T2D3	T2D4	HT total	T2D total	p-value
		scRNA	Cell #	166	362	267	146	403	525	955	130	527	476	844	
	HNF1A + cell #	0	2	0	1	6	1	6	1	0	0	4	16	5	
snATAC	Cell #	2,272	277	164	313	981	649	3,012	391	1,396	1,683	2,409	7,668	5,879	1.8e-09
	HNF1A + cell #	150	8	2	18	77	50	145	13	27	39	134	450	213	

5. Although the authors predicted the target genes of HNF1A, they did not perform HNF1A ChIP-seq to verify the target genes.

We are happy to report that we have now successfully performed HNF1A Cut&Run in EndoC- β H3 cells (**Reviewer Fig 4** below and the new **Figure 7j** and **Supplementary Fig 6**). We made a lot of efforts obtaining this new data including testing multiple antibodies and protocols. We see HNF1A binding at nearly all the predicted target genes including *FXVD2* (**Reviewer Fig 4**). Furthermore, most of the HNF1A binding colocalizes with ATAC-seq peaks that are down-regulated in the β -cells from T2D-donors.

Reviewer Fig 4. HNF1A Cut&Run in EndoC- β H3 cells. Genome browser snapshot of *FXVD2* gene locus. HNF1A Cut&Run track is in blue. Green tracks are pseudo-bulk ATAC-seq data after pooling β -cells from healthy or T2D donors.

6. Can the authors apply RePACT on HiC dataset to find differential loops? Do the loop between HNF1A and its CRE change following “intra-donor heterogenous”?

We thank the reviewer for the suggestion. Unfortunately, we cannot perform RePACT on Hi-C data because RePACT is designed to analyze single cell data. The Hi-C experiments in this study are bulk Hi-C performed on sorted α - or β -cells.

Minor points:

1. Line 129 cited Fig. 3b. Should be Fig. 3d.

We thank the reviewer for pointing this out and have corrected the citation.

2. Starting from line 228, HNF-1 α should be changed to HNF1A, consistent with the previous text.

We have updated the text as the reviewer suggested.

Reviewer #3 (Remarks to the Author):

Weng et al. present a comprehensive multi-omic study of islets from healthy and T2D donors, including scRNA-seq, snATAC-seq, and Hi-C. The authors extend their previous

RePACT method to snATAC-seq and show beta cell heterogeneity at the epigenomic level, in addition to the transcriptomic level. Importantly, the authors identify HNF1A as a putative driver for beta cell heterogeneity through the integration of multi-omic datasets and infer its function through the HNF1A-FXYD2 pathway. Overall, the study is well done, and the figures are of high quality. However, there are a few concerns that need to be addressed.

We thank the reviewer for the supportive comments.

1. As far as I noticed, basic QC metrics such as the number of reads and the number of genes/peaks per cell for scRNA-seq and snATAC-seq data are not provided. Including these statistics would provide a better understanding of the generated dataset.

We have added QC metrics to **Supplementary_Data5.xls**.

2. While the discrepancy between cell-type-specific ATAC peaks and compartment switches is intriguing, the 3D genome part of the manuscript seems somewhat unrelated to the overall story, except for the loops being used to link ATAC-seq peaks to genes. Additionally, the list of loops should be provided as a supplementary table, as it can be a valuable resource for the field.

In this revision, we moved some of the Hi-C figure panels to supplementary figures and shortened some text to improve the coherence of the paper.

As the reviewer pointed out, we performed Hi-C as a part of the multiomic dataset to connect epigenome (ATAC peaks) to transcriptome. The comparison between α - and β -cells proves that there is a significant association between chromatin loops, enhancers, and gene expression.

We have added the list of loops to **Supplementary_Data6.xls**.

3. For Figure 4d, while the heatmap suggests that this study has similar gene patterns with Fang et al., there is no quantitative measure to show the consistency.

We have added a new analysis examining the overlap of “intra-donor” and “inter-donor” genes to the genes called from Fang *et al.* (**Reviewer Fig. 1** below and the new **Fig. 4e** in the revision). We found that both “intra-donor” and “inter-donor” genes are significantly enriched with the marker genes called by Fang *et al.*

Reviewer Fig. 1. Reproducibility between this study and Fang et al.

4. For the peak-to-gene assignment in Figure 6, it is unclear whether the peak-to-gene relationship is always one-to-one. If there are multi-to-multi assignments, how they add up for the T2D trajectory is unclear

Figure 6a shows peak-to-gene assignments where both peak and genes are down-regulated along the T2D trajectory. The orange/green lines denoting peak-to-gene assignments based on promoter proximity or Hi-C connection. The purpose of this analysis is to find possible epigenetic events that cause transcription changes. It can be seen from the figures that nearly all peaks connect to only one gene; some genes do have multiple supporting peaks. In these figures, we ignored the unchanged peaks that are connected to dynamic genes since those peaks cannot explain the expression data. There is no conflicting peak-gene links, *i.e.*, out of the 69 down-regulated genes in **Fig. 6a**, none of them have links to up-regulated peaks.

5. While it is nice to have the RePACT code provided, there are no codes to replicate major results of the paper. It is critical to have these codes for reproducibility.

We have added example data and source codes to GitHub (https://github.com/JinLabBioinfo/RePACT/tree/master/2023_paper_code).

REVIEWERS' COMMENTS

Reviewer #1 (Remarks to the Author):

The authors have addressed the comments raised in my previous review. Specifically, they validated the results of the RePact model with a continuous disease metric (HbA1C) instead of a dichotomous label to find T2D associated genes. They also checked the reproducibility of their results with those obtained in a previous report from the same group. I think that this strengthens the conclusions of the paper, and I support its publication.

Reviewer #2 (Remarks to the Author):

The authors provide satisfactory answers to our comments.

Reviewer #3 (Remarks to the Author):

The authors have responded to all my concerns thoroughly, and I don't have additional concerns. I believe the study's findings and the data generated would be valuable resources in the field.